# RETHINKING THE GOLD STANDARD:
# WHY DISCRETE CURVATURE FAILS TO FULLY
# CAPTURE OVER-SQUASHING IN GNNS?

**Jialong Chen**[1]**, Bowen Deng, Zibin Zheng, Chuan Chen**[2,*]
Sun Yat-sen University
[1]`chenjlong7@mail2.sysu.edu.cn`, [2]`chenchuan@mail.sysu.edu.cn`

## ABSTRACT

As a topological invariant for discrete structures, discrete curvature has been widely adopted in the study of complex networks and graph neural networks. A prevailing viewpoint posits that edges with highly negative curvature will induce graph bottlenecks and the over-squashing phenomenon. In this paper, we critically re-examine this view and put forward our central claim: **high negative curvature is a sufficient but not a necessary condition for over-squashing**. We first construct a family of counterexamples demonstrating the failure of discrete curvature, where some edges are severely squashed, but the curvature still appears positive. Furthermore, extensive experiments demonstrate that the most commonly used discrete curvature measure — Ollivier–Ricci curvature — fails to detect as many as $30\% \sim 40\%$ of over-squashed edges. To alleviate this limitation, we propose Weighted Augmented Forman-3 Curvature (WAF3), which significantly improves the detection of over-squashed edges. Additionally, we develop a highly efficient approximation algorithm for WAF3, enabling curvature computation on graphs with five million edges in only 23.6 seconds, which is 133.7 times faster than the existing algorithm with the lowest complexity for curvatures.

## 1 INTRODUCTION

In differential geometry, curvature is used to describe how volume grows within a local region and how geodesics diverge or converge. Discrete curvature naturally extends this concept to discrete structures, such as graphs. In complex network analysis and graph deep learning, discrete curvature plays a critical role. It is widely applied in numerous downstream tasks, including key node identification (Farooq et al., 2019), community detection (Park & Li, 2024; Sia et al., 2019), clustering (Tian et al., 2025; Sun et al., 2023), sparsification (Zhang et al., 2023), and anomaly detection (Grover et al., 2025), among others. Among these, the idea of tackling the graph over-squashing (Akansha, 2025) based on discrete curvature has been widely studied, with one of the most notable findings being that:

> *"Edges with high negative curvature are those causing the graph bottleneck and thus leading to the over-squashing phenomenon."* — Topping et al. (2021)

This perspective has garnered significant attention within the community. It has spawned a considerable body of follow-up research, including theoretical investigations into the relationship between over-squashing and curvature (Di Giovanni et al., 2023a; Nguyen et al., 2023), curvature-based graph rewiring techniques (Nguyen et al., 2023; Giraldo et al., 2023; Fesser & Weber, 2024), and curvature-inspired graph neural networks (Li et al., 2022; Sun et al., 2022; Fu et al., 2025).

Although the work of Topping et al. (2021) has achieved remarkable success, a subtle but important point is that they only established the sufficiency of highly negative curvature for over-squashing, while leaving its necessity unaddressed. To the best of our knowledge, this fundamental yet crucial issue has long been overlooked. In this paper, we investigate it for the first time and uncover a striking fact: the necessity does not actually hold. In other words, there exist edges in graph datasets that suffer from severe over-squashing but cannot be detected by curvature.

---

*Corresponding author.

In detail, our contributions are organized into four interlocking parts:

❶ **New theoretical results (Section 3)**. We construct a family of counterexample graphs and prove that within them, there exist edges that, despite exhibiting severe over-squashing, still possess highly positive discrete curvature. Here, discrete curvature can be defined in eight popular ways, including Ollivier Ricci curvature (Ollivier, 2009) and Balanced Forman curvature (Topping et al., 2021), among others. Theorem 4 implies that curvature may ignore some over-squashed edges, so highly negative curvature is not a necessary condition for over-squashing.

❷ **New metric and extensive empirical evidence (Section 4)**. To answer the question of how many over-squashed edges in practical graph learning tasks are overlooked by curvature, we propose a new metric: **M**issed **O**ver-**S**quashing **R**atio (MOSR). MOSR quantifies the proportion of over-squashed edges that are ignored by curvature. Extensive experimental results show that the MOSR of Ollivier Ricci curvature can exceed 30%, revealing a significant deficiency in one of the most widely used discrete curvature measures. Further experimental analysis provides insights into the underlying reasons why curvature fails to capture these edges.

❸ **New discrete curvature (Section 5)**. Based on the aforementioned analysis and experimental observations, we propose a new discrete curvature definition called WAF3. Not only does WAF3 achieve a significantly lower MOSR value compared to other existing curvature definitions, but it also maintains computational complexity on par with the most efficient one currently available.

❹ **New approximation algorithm (Section 6)**. Although WAF3 already boasts state-of-the-art computational complexity, it still faces scalability limitations when processing large-scale graphs. To address this, we propose an efficient approximation algorithm for WAF3 that further reduces the time complexity to a linear level. This algorithm requires only 23.8 seconds to complete computations on a graph containing five million edges, achieving a 133.7× speedup compared to exact computation.

In summary, our work is not only critical, demonstrating through both theory and experiments that curvature cannot fully capture the over-squashing in GNNs, but also constructive: we propose new metrics, curvature definitions, and approximation algorithms, significantly enhancing the applicability of curvature-based tools in graph learning.

## 2 PRELIMINARY

**Message passing neural network (MPNN, (Gilmer et al., 2017; Kipf, 2016))** Consider a simple, connected, undirected graph $\mathcal{G} = (\mathcal{V}, \mathcal{E})$ with a node set of $\mathcal{V}$ and an edge set $\mathcal{E}$. We denote the adjacency matrix of $\mathcal{G}$ by $\mathbf{A}$. Each node is equipped with an initial feature vector $\mathbf{h}^{(0)} \in \mathbb{R}^{d_0}$. $\{\mathbf{W}^{(l)} \in \mathbb{R}^{d_l \times d_{l+1}}\}_l$ is a set of learnable parameters. The $l$-th layer of MPNN is formalized as:

$$\mathbf{H}^{(l+1)} = \mathsf{ReLU}\left(\widetilde{\mathbf{A}}\mathbf{H}^{(l)}\mathbf{W}^{(l)}\right). \tag{1}$$

where $\widetilde{\mathbf{A}} := (\mathbf{D} + \mathbf{I})^{-1/2}(\mathbf{A} + \mathbf{I})(\mathbf{D} + \mathbf{I})^{-1/2}$ denotes the symmetrically normalized adjacency matrix and $\mathbf{D}$ denotes the degree matrix. $\mathbf{H}^l := [\mathbf{h}_0^{(l)}; \cdots ; \mathbf{h}_{|\mathcal{N}|}^{(l)}]^T \in \mathbb{R}^{|\mathcal{N}| \times d_l}$ denotes the collections of $l$-layer embedding of all nodes. In line with many previous works, we also introduce the following assumptions to facilitate the analysis of the ReLU.

**Assumption 1.** *(Di Giovanni et al., 2023a; Kawaguchi, 2016; Xu et al., 2018) All paths in the computation graph of the model are activated with the same probability of success $\rho$.*

**Discrete curvatures** For any edge $u \sim v$ in graph $\mathcal{G}$, the edge curvature $\mathsf{Curv}(u, v)$ measures the tightness of the connection between the first-order ego-graph of node $u$ and the first-order ego-graph of node $v$. As shown in Table 1, there are multiple definitions for $\mathsf{Curv}$. Regardless of the specific definition, discrete curvature typically depends only on a tiny local neighborhood of the edge $u \sim v$ (except for the resistance curvature (Devriendt & Lambiotte, 2022)).

**Over-squashing** In graph deep learning, message passing often compresses information from large neighborhoods into single topological structures (e.g., nodes or edges), resulting in information loss and gradient issues — known as over-squashing (Alon & Yahav, 2020), which has emerged as a key challenge in modern graph models. The Jacobian norm between the feature of the input node and the embedding of the output node offers the most accurate measure of information flow, and thus the

Table 1: We summarize all discrete curvatures defined on edges here. Curvatures defined on nodes (such as Bakry-Émery-Ricci (Mondal et al., 2024), combination (Kamtue, 2018), and node resistance (Devriendt & Lambiotte, 2022)) are not included. $\mu_u^\alpha$ is the uniform distribution of the first-order neighbors of $u$ with restart probability $\alpha$. $W_1$ is the 1-Wasserstein distance. $\{w_{uv}\}$ is the pseudo-inverse of the weighted Laplacian matrix. $d_u$ is the degree of node $u$, and $d_u \vee d_v := \max(d_u, d_v)$, $d_u \wedge d_v := \min(d_u, d_v)$. $\triangle(u,v)$ and $\square(u,v)$ denote the number of triangles and quadrangles containing the edge $(u,v)$. $\#_\square^u(u,v)$ denote the numbers of neighbors of $u$ forming a 4-cycle based at the edge $(u,v)$ without diagonals inside. $\gamma(u,v)$ is the maximal number of 4-cycles based at $(u,v)$ traversing a common node. $f$ is the degree-weighting function.

| Curvature | Definition | Complexity |
|---|---|---|
| Ollivier ricci (Ollivier, 2009) | $1 - W_1(\mu_u, \mu_v)$ | $\mathcal{O}(\lvert\mathcal{E}\rvert d_{\max}^3)$ |
| Lin-Lu-Yau (Lin et al., 2011) | $\lim_{\alpha \to 1^-} (1 - W_1(\mu_u^\alpha, \mu_v^\alpha))/(1-\alpha)$ | $\mathcal{O}(\lvert\mathcal{E}\rvert d_{\max}^3)$ |
| Link resistance (Devriendt & Lambiotte, 2022) | $(2 - \sum_{i \sim u} w_{ui} - \sum_{j \sim v} w_{vj})/w_{uv}$ | $\mathcal{O}(\lvert\mathcal{V}\rvert^3)$ |
| Balance forman (Topping et al., 2021) | $\frac{2}{d_u} + \frac{2}{d_v} - 2 + 2\frac{\triangle(u,v)}{d_u \vee d_v} + \frac{\triangle(u,v)}{d_u \wedge d_v} + \frac{\#_\square^u(u,v) + \#_\square^v(u,v)}{\gamma(u,v)(d_u \vee d_v)}$ | $\mathcal{O}(\lvert\mathcal{E}\rvert d_{\max}^2)$ |
| Balance Forman w/o 4-cycle (Tori et al., 2024a) | $\frac{2}{d_u} + \frac{2}{d_v} - 2 + 2\frac{\triangle(u,v)}{d_u \vee d_v} + \frac{\triangle(u,v)}{d_u \wedge d_v}$ | $\mathcal{O}(\lvert\mathcal{E}\rvert d_{\max})$ |
| Jost-Liu Forman (Jost & Liu, 2014) | $-(1 - \frac{1}{d_u} - \frac{1}{d_v} - \frac{\triangle(u,v)}{d_u \vee d_v})_+ - (1 - \frac{1}{d_u} - \frac{1}{d_v} - \frac{\triangle(u,v)}{d_u \wedge d_v})_+ + \frac{\triangle(u,v)}{d_u \vee d_v}$ | $\mathcal{O}(\lvert\mathcal{E}\rvert d_{\max})$ |
| Augmented Forman-3 (Forman, 2003) | $4 - d_u - d_v + 3\triangle(u,v)$ | $\mathcal{O}(\lvert\mathcal{E}\rvert d_{\max})$ |
| Augmented Forman-4 (Forman, 2003) | $4 - d_u - d_v + 3\triangle(u,v) + 2\square(u,v)$ | $\mathcal{O}(\lvert\mathcal{E}\rvert d_{\max}^2)$ |
| Weighted AF-3 (Ours) | $\sum_{i \in \mathcal{B}(u) \cap \mathcal{B}(v)} f(d_i) - \left( \sum_{i \in \mathcal{N}(u)/\mathcal{B}(v)} f(d_i) + \sum_{i \in \mathcal{N}(v)/\mathcal{B}(u)} f(d_i) \right)$ | $\mathcal{O}(\lvert\mathcal{E}\rvert d_{\max})$ $\mathcal{O}(H\lvert\mathcal{E}\rvert)$ |

severity of over-squashing. In Lemma 2, we establish a lower bound for the Jacobian norm when the degrees of the two ends are fixed at $a$ and $b$. All proofs are provided in Appendix A.

**Lemma 2** (Infimum of the over-squashing). *Let $N, a, b$ be positive integers. The set $\mathbb{G}_N((\mathsf{s}, a), (\mathsf{t}, b))$ denotes the collection of all simple undirected graphs $\mathcal{G} = (\mathcal{V}, \mathcal{E})$ with $\lvert\mathcal{V}\rvert = N$ such that there exist adjacent vertices $\mathsf{s}, \mathsf{t} \in \mathcal{V}$ where $\deg(\mathsf{s}) = a$ and $\deg(\mathsf{t}) = b$. Assume an $L$-layer MPNN as in equation (1). Then:*

$$\inf_{\substack{N \in \mathbb{Z}^+ \\ \mathcal{G} \in \mathbb{G}_N((\mathsf{s},a),(\mathsf{t},b))}} \left\| \frac{\partial \mathbf{h}_\mathsf{t}^{(L)}}{\partial \mathbf{h}_\mathsf{s}^{(0)}} \right\| = \phi_L(a, b),$$

*where $\phi_L(a, b) := \left\| \prod_{l=0}^{L-1} \mathbf{W}^{(l)} \right\| \left( \frac{1}{a+1} + \frac{1}{b+1} \right)^{L-1} \frac{\rho}{\sqrt{(a+1)(b+1)}}$.*

## 3 DISCRETE CURVATURES FAIL TO FULLY CAPTURES OVER-SQUASHING

Since the Jacobian matrix norm is tightly coupled with the model and entails prohibitive computational costs, Topping et al. (2021) first proposed using discrete curvature as a surrogate criterion for detecting over-squashing. However, through the following counterexamples and theorems, we demonstrate that discrete curvature is in fact not a necessary condition for identifying over-squashing.

**Definition 3** (counterexample graph). *Let $\mathsf{s}$ the soruce node, $\mathsf{t}$ the target node, $\mathcal{N}_1 = \{u_i\}_{1 \le i \le n}$ the 1-hop neighbor set and $\mathcal{N}_2 = \{v_{ij}\}_{1 \le i \le n, 1 \le j \le m}$ the 2-hop neighbor set. The counterexample graph $\mathcal{G}_{n,m}^\mathsf{c}$ is such a simple, connected and undirected graph that $\mathcal{V} := \{\mathsf{s}, \mathsf{t}\} \cup \mathcal{N}_1 \cup \mathcal{N}_2$ and $\mathcal{E} := \{(\mathsf{s}, \mathsf{t})\} \cup \mathcal{E}_1 \cup \mathcal{E}_2$, which $\mathcal{E}_1 := \{(\mathsf{s}, u_i)\}_{1 \le i \le n} \cup \{(\mathsf{t}, u_i)\}_{1 \le i \le n}$ and $\mathcal{E}_2 := \{(u_i, v_{ij})\}_{1 \le i \le n, 1 \le j \le m}$.*

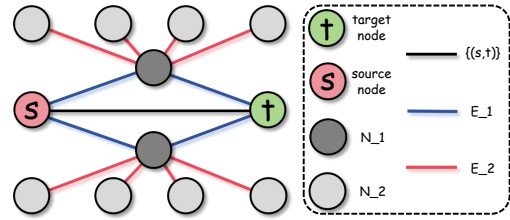

Figure 1: Example diagram of $\mathcal{G}_{2,4}^\mathsf{c}$.

> **Theorem 4** (Discrete curvatures fail to fully to capture over-squashing). *Consider an L-layer GCN as in equation (1). For the family of graphs $\mathscr{G} = \{\mathcal{G}^c_{n,m}\}_{n,m \in \mathbb{R}^+}$ with source-target pairs $(\mathsf{s}, \mathsf{t})$, we have:*
>
> - *For any fixed $n^*$, $\left\| \frac{\partial \mathbf{h}^{(L)}_{\mathsf{t}}}{\partial \mathbf{h}^{(0)}_{\mathsf{s}}} \right\| \to \phi_L(n^*+1, n^*+1)$ at speed $\mathcal{O}(m^{-1})$ when $m \to +\infty$.*
> - *If $\mathsf{Curv}$ is given by any of: $\alpha$-Ollivier-Ricci curvature, Lin-Lu-Yau curvature, balanced Forman curvature (with/without 4-cycle), Jost-Liu curvature or augmented Forman-3/4 curvature, then there exist a $c > 0$, such that $\forall \mathcal{G} \in \mathscr{G}$, $\mathsf{Curv}(\mathsf{s}, \mathsf{t}) > c$.*
> - *If $\mathsf{Curv}$ is defined as resistance curvature under symmetric normalized Laplacian, then for any fixed $n^*$, there exist a $m^*$ and a $c > 0$, such that $\forall \mathcal{G} \in \{\mathcal{G}^c_{n,m} | n = n^*, m > m^*\}$, $\mathsf{Curv}(\mathsf{s}, \mathsf{t}) > c$.*

In simple terms, the above theorem states that in $\mathcal{G}^c_{n,m}$, as long as $m$ is large enough, there will be very severe over-squashing between source node $\mathsf{s}$ and target node $\mathsf{t}$. However, all discrete curvatures fail to identify this — their values are all positive.

To understand this phenomenon, we need to keep in mind that the discrete curvature between $(\mathsf{s}, \mathsf{t})$ is essentially a measure of how tightly connected the first-order neighbors of $\mathsf{s}$ are to the first-order neighbors of $\mathsf{t}$ (Chen et al., 2025). Furthermore, this "tightness" can usually be reflected in the number of triangles that $(\mathsf{s}, \mathsf{t})$ participates in (Topping et al., 2021; Jost & Liu, 2014; Forman, 2003). Since any edge $(u, v)$ can form at most $\min\{d_u - 1, d_v - 1\}$ triangles, and in $\mathcal{G}^c_{n,m}$, $(\mathsf{s}, \mathsf{t})$ just reaches this upper limit ($d_\mathsf{s} = d_\mathsf{t} = n + 1$ and there are $n$ triangles: $\mathsf{s} - u_i - \mathsf{t}, \forall i \in [1, n]$), $\mathsf{Curv}(\mathsf{s}, \mathsf{t})$ under any definition is almost always highly positive.

On the other hand, in $\mathcal{G}^c_{n,m}$, when the information in node $\mathsf{s}$ propagates to node $\mathsf{t}$, a considerable part of the propagation path needs to pass through the nodes in $\mathcal{N}_1$. However, each node in $\mathcal{N}_1$ is connected to $m$ other nodes. Every time the propagation path passes through a node in $\mathcal{N}_1$, the information in $\mathsf{s}$ is "diluted" by $m$ nodes. As $m$ increases, the degree of over-squashing between $\mathsf{s}$ and $\mathsf{t}$ approaches its theoretical lower bound. Since almost all GNNs require at least two rounds of message-passing, discrete curvature that only considers first-order neighbors is almost always insufficient to capture all potential over-squashing fully.

## 4 THE MISSED OVER-SQUASHING EDGES: HOW MANY AND WHERE

Since discrete curvature cannot fully identify over-squashing in theory, two other important questions arise: *how many over-squashing edges are ignored by curvature in practice? Moreover, where are these ignored edges?*

To answer the first question, we first establish corresponding metrics. For any graph $\mathcal{G} = (\mathcal{V}, \mathcal{E})$, we define the multiset $\mathcal{C} := \{\{\mathsf{Curv}(e) | e \in \mathcal{E}\}\}$ as the collection of discrete curvature values for all edges in $\mathcal{E}$. Let $\mathcal{C}_- := \{\{c | c < 0, c \in \mathcal{C}\}\}$ be the subset of negative curvature values. The function $\mathsf{Percentile}(\mathcal{C}_-, q)$ returns the value below which $q\ \%$ of the observations in $\mathcal{C}_-$ lie, for $q \in [0, 100]$. We then define:

$$\mathcal{E}_q := \{e \in \mathcal{E} | \mathsf{Curv}(e) \le \mathsf{Percentile}(\mathcal{C}_-, q)\}. \tag{2}$$

Note that $\mathcal{E}_q$ corresponds precisely to the set of "edges with high negative curvature" mentioned by Topping et al. (2021). That is, the edges in $\mathcal{E}_q$ are those correctly identified as over-squashing by curvature. Here, $q$ represents the threshold for classifying an edge as over-squashing; a smaller $q$ implies a stricter criterion.

For an $L$-layer message passing neural network, let $\mathsf{JacoNorm}(u, v) := \|\partial \mathbf{h}^{(L)}_v / \mathbf{h}^{(0)}_u\|_F$ denote the ground-truth measure of information squashing. Define $J_q := \max_{(u,v) \in \mathcal{E}_q} \mathsf{JacoNorm}(u, v)$. We then introduce the **M**issed **O**ver-**S**quashing **R**atio (MOSR):

$$\mathsf{MOSR}_q := \frac{\sum_{(u,v) \in \mathcal{E}} \mathbf{1}_{\mathsf{Curv}(u,v) \ge 0} \cdot \mathbf{1}_{\mathsf{JacoNorm}(u,v) \le J_q}}{\sum_{(u',v') \in \mathcal{E}} \mathbf{1}_{\mathsf{JacoNorm}(u',v') \le J_q}}. \tag{3}$$

In $\mathsf{MOSR}_q$, the numerator counts edges with non-negative curvature that are nevertheless over-squashed (i.e., $\mathsf{JacoNorm}(u, v) \le J_q$), which are thus ignored by curvature. The denominator counts all truly over-squashing edges (with $J_q$ as the threshold). Therefore, $\mathsf{MOSR}_q$ **represents the proportion of over-squashing edges that are not identified by curvature**.

Table 2: The values of $MOSR_{10}$ and $MOSR_{25}$ across different GNNs, curvatures, and datasets. Among these, the entry ".030/.103" in the first row and first column indicates that for Ollivier Ricci curvature, GCN, and Cora dataset, $MOSR_{10} = 0.030$ and $MOSR_{25} = 0.103$. OOR denotes "Out of Resources", meaning the GPU memory consumption exceeds 24 GB or the running time surpasses 12 hours. NNE (No Negative-curvature Edge) indicates that $|\mathcal{E}_q| = 0$ in this scenario.

| | Ollivier Ricci | | | Augmented Forman-3 | | | Balanced Forman | | |
|---|---|---|---|---|---|---|---|---|---|
| | GCN | GAT | SAGE | GCN | GAT | SAGE | GCN | GAT | SAGE |
| **Cora** | .030/.103 | .185/.224 | .233/.258 | .001/.027 | .163/.170 | .187/.187 | .002/.010 | .155/.217 | .210/.245 |
| **Citeseer** | .119/.151 | .172/.277 | .286/.351 | .008/.034 | .239/.239 | .289/.307 | .000/.014 | .087/.120 | .219/.324 |
| **Pubmed** | .026/.073 | .087/.090 | .097/.097 | .000/.001 | .009/.009 | .011/.011 | .000/.000 | .074/.108 | .144/.142 |
| **Computers** | .352/.352 | .353/.353 | .353/.353 | .003/.006 | .011/.011 | .011/.011 | .000/.005 | .010/.012 | .011/.012 |
| **Photo** | .503/.503 | .503/.503 | .503/.503 | .042/.043 | .045/.045 | .045/.045 | .041/.044 | .045/.046 | .045/.046 |
| **CS** | .057/.086 | .113/.128 | .125/.126 | .017/.030 | .074/.075 | .075/.075 | .001/.012 | .041/.073 | .080/.091 |
| **Physics** | OOR | OOR | OOR | .006/.027 | .052/.056 | OOR | .001/.012 | .055/.065 | OOR |
| **WikiCS** | .472/.473 | .473/.473 | .473/.473 | .250/.249 | .249/.249 | .249/.249 | .245/.244 | .246/.245 | .250/.246 |
| **Cora_ML** | .204/.204 | .217/.235 | .210/.225 | .025/.039 | .096/.098 | .099/.101 | .056/.109 | .126/.139 | .126/.147 |
| **Cora_Full** | OOR | OOR | OOR | OOR | OOR | OOR | OOR | OOR | OOR |
| **DBLP** | .072/.080 | .084/.089 | .086/.096 | .003/.004 | .030/.033 | .036/.038 | .007/.017 | .038/.052 | .048/.065 |
| **Cornell** | .000/.000 | .383/.395 | .396/.410 | .000/.000 | .136/.138 | .167/.158 | .000/.364 | .335/.370 | .373/.390 |
| **Texas** | .093/.174 | .308/.308 | .308/.308 | .000/.000 | .112/.116 | .126/.130 | .000/.254 | .385/.370 | .342/.366 |
| **Wisconsin** | .000/.246 | .334/.334 | .332/.342 | .000/.000 | .113/.118 | .128/.135 | .175/.191 | .257/.256 | .221/.260 |
| **Chameleon** | .643/.643 | .643/.643 | .643/.643 | .115/.116 | .125/.125 | .129/.129 | .010/.120 | .131/.135 | .136/.136 |
| **Squirrel** | .723/.723 | .723/.723 | .723/.723 | .137/.137 | .133/.133 | .133/.133 | .130/.129 | .130/.130 | .132/.132 |
| **Roman-empire** | .014/.097 | .276/.446 | .497/.547 | .001/.080 | .430/.520 | .524/.524 | .000/.000 | .247/.532 | .695/.753 |
| **Tolokers** | .657/.657 | OOR | .657/.657 | .002/.002 | OOR | .003/.003 | .003/.003 | OOR | .003/.003 |
| **Questions** | OOR | OOR | OOR | .000/.002 | .008/.008 | .010/.010 | .048/.074 | .083/.085 | .103/.100 |
| **Amazon-ratings** | .639/.655 | .524/.638 | .638/.653 | .228/.285 | .292/.297 | .309/.310 | .335/.378 | .117/.275 | .357/.357 |
| **Minesweeper** | NNE | NNE | NNE | .502/.502 | .502/.502 | .502/.502 | .502/.502 | .502/.502 | .502/.502 |
| **Average** | **.271/.307** | **.336/.366** | **.386/.398** | **.067/.079** | **.148/.155** | **.160/.161** | **.078/.124** | **.161/.196** | **.210/.227** |

As shown in Table 2, we comprehensively report the results of $MOSR_{10}$ and $MOSR_{25}$ along with their average values across three of the most commonly used curvatures, three of the most widely adopted graph neural networks, and 21 datasets, amounting to a total of 350 numerical results.

- **Observation 1**: At $q = 10$, discrete curvature was systematically ignored $6.7\% \sim 38.6\%$ of over-squashing edges. When $q$ increased to 25, this range increased to $7.9\% \sim 39.8\%$. This indicates that discrete curvature fails to identify over-squashing phenomena perfectly.

- **Observation 2**: Across different datasets, discrete curvature generally demonstrates superior performance on GAT compared to GraphSAGE, while GCN consistently achieves the optimal results. This indicates that model architecture significantly influences the accurate identification of over-squashing edges.

- **Observation 3**: The Ollivier Ricci curvature, with its computational complexity as high as $\mathcal{O}(|\mathcal{E}|d_{\max}^3)$, missed the most significant number of over-squashing edges, whereas the Augmented Forman-3 curvature — the one with the lowest complexity — achieved the best average performance. Therefore, we recommend prioritizing the latter in GNNs.

To answer the second question, we introduce edge betweenness (Freeman, 1977; Girvan & Newman, 2002). According to Girvan & Newman (2002), the edge betweenness of an edge $e$ is defined as:

$$\text{Between}(e) = \sum_{u \neq v \in \mathcal{V}} \frac{\sigma_{uv}(e)}{\sigma_{uv}}. \tag{4}$$

Where $\sigma_{uv}$ denotes the total number of shortest paths between nodes $u$ and $v$; $\sigma_{uv}(e)$ denotes the number of those shortest paths that pass through edge $e$. Simply put, **a high** $\text{Between}(e)$ **usually means that edge** $e$ **forms a bridge between two clusters, and a low** $\text{Between}(e)$ **indicates that** $e$ **is inside a cluster**. We further introduce the following three statistics to characterize the average edge

Table 3: The model is fixed as GCN, $q$ is set to 25, and we report BetwIden, BetwAll, and BetwIgno on three curvature definitions and 21 datasets, respectively. NIE (No Ignored Edge) means no edges are ignored by curvature, so BetwIgno cannot be calculated.

| | Ollivier Ricci | | | Augmented Forman-3 | | | Balanced Forman | | |
|---|---|---|---|---|---|---|---|---|---|
| | BetwIden | BetwAll | BetwIgno | BetwIden | BetwAll | BetwIgno | BetwIden | BetwAll | BetwIgno |
| **Cora** | $1.04 \times 10^4$ | $3.69 \times 10^3$ | $7.58 \times 10^2$ | $8.41 \times 10^3$ | $3.69 \times 10^3$ | $4.75 \times 10^2$ | $9.91 \times 10^3$ | $3.69 \times 10^3$ | $2.36 \times 10^3$ |
| **Citeseer** | $1.75 \times 10^4$ | $4.60 \times 10^3$ | $1.02 \times 10^3$ | $9.44 \times 10^3$ | $4.60 \times 10^3$ | $5.06 \times 10^2$ | $1.16 \times 10^4$ | $4.60 \times 10^3$ | $1.09 \times 10^3$ |
| **Pubmed** | $6.75 \times 10^4$ | $2.78 \times 10^4$ | $1.55 \times 10^4$ | $4.74 \times 10^4$ | $2.78 \times 10^4$ | $2.62 \times 10^3$ | $5.93 \times 10^4$ | $2.78 \times 10^4$ | NIE |
| **Computers** | $4.32 \times 10^3$ | $1.23 \times 10^3$ | $2.98 \times 10^2$ | $2.44 \times 10^3$ | $1.23 \times 10^3$ | $1.79 \times 10^2$ | $2.19 \times 10^3$ | $1.23 \times 10^3$ | $2.70 \times 10^2$ |
| **Photo** | $4.02 \times 10^3$ | $9.53 \times 10^2$ | $2.81 \times 10^2$ | $1.50 \times 10^3$ | $9.53 \times 10^2$ | $2.36 \times 10^2$ | $1.91 \times 10^3$ | $9.53 \times 10^2$ | $3.31 \times 10^2$ |
| **CS** | $2.54 \times 10^4$ | $1.11 \times 10^4$ | $2.19 \times 10^3$ | $2.22 \times 10^4$ | $1.11 \times 10^4$ | $1.39 \times 10^3$ | $2.35 \times 10^4$ | $1.11 \times 10^4$ | $1.54 \times 10^3$ |
| **Physics** | OOR | OOR | OOR | $1.86 \times 10^4$ | $1.24 \times 10^4$ | $1.43 \times 10^3$ | $2.30 \times 10^4$ | $1.24 \times 10^4$ | $1.03 \times 10^3$ |
| **WikiCS** | $3.36 \times 10^3$ | $8.94 \times 10^2$ | $1.77 \times 10^2$ | $3.01 \times 10^3$ | $8.94 \times 10^2$ | $7.94 \times 10^1$ | $1.82 \times 10^3$ | $8.94 \times 10^2$ | $1.81 \times 10^2$ |
| **Cora_ML** | $7.10 \times 10^3$ | $2.55 \times 10^3$ | $7.96 \times 10^2$ | $5.24 \times 10^3$ | $2.55 \times 10^3$ | $2.54 \times 10^2$ | $5.64 \times 10^3$ | $2.55 \times 10^3$ | $1.24 \times 10^3$ |
| **Cora_Full** | OOR | OOR | OOR | OOR | OOR | OOR | OOR | OOR | OOR |
| **DBLP** | $3.79 \times 10^4$ | $1.64 \times 10^4$ | $4.53 \times 10^3$ | $2.56 \times 10^4$ | $1.64 \times 10^4$ | $1.02 \times 10^3$ | $2.89 \times 10^4$ | $1.64 \times 10^4$ | $9.19 \times 10^3$ |
| **Cornell** | $5.37 \times 10^2$ | $1.92 \times 10^2$ | NIE | $3.30 \times 10^2$ | $1.92 \times 10^2$ | NIE | $4.71 \times 10^2$ | $1.92 \times 10^2$ | $1.82 \times 10^2$ |
| **Texas** | $4.08 \times 10^2$ | $1.81 \times 10^2$ | $8.45 \times 10^1$ | $3.10 \times 10^2$ | $1.81 \times 10^2$ | NIE | $3.44 \times 10^2$ | $1.81 \times 10^2$ | $1.82 \times 10^2$ |
| **Wisconsin** | $5.93 \times 10^2$ | $2.27 \times 10^2$ | $1.32 \times 10^2$ | $4.65 \times 10^2$ | $2.27 \times 10^2$ | NIE | $4.21 \times 10^2$ | $2.27 \times 10^2$ | $1.17 \times 10^2$ |
| **Chameleon** | $1.66 \times 10^3$ | $2.94 \times 10^2$ | $9.02 \times 10^1$ | $7.25 \times 10^2$ | $2.94 \times 10^2$ | $3.45 \times 10^1$ | $4.72 \times 10^2$ | $2.94 \times 10^2$ | $6.64 \times 10^1$ |
| **Squirrel** | $1.63 \times 10^3$ | $2.31 \times 10^2$ | $6.54 \times 10^1$ | $3.40 \times 10^2$ | $2.31 \times 10^2$ | $4.03 \times 10^1$ | $5.44 \times 10^2$ | $2.31 \times 10^2$ | $6.95 \times 10^1$ |
| **Roman-empire** | $5.28 \times 10^7$ | $1.82 \times 10^7$ | $7.48 \times 10^5$ | $3.54 \times 10^7$ | $1.82 \times 10^7$ | $1.90 \times 10^6$ | $4.18 \times 10^7$ | $1.82 \times 10^7$ | NIE |
| **Tolokers** | $1.64 \times 10^3$ | $3.71 \times 10^2$ | $1.39 \times 10^2$ | $7.76 \times 10^2$ | $3.71 \times 10^2$ | $3.15 \times 10^1$ | $4.71 \times 10^2$ | $3.71 \times 10^2$ | $2.31 \times 10^3$ |
| **Questions** | OOR | OOR | OOR | $5.37 \times 10^4$ | $3.34 \times 10^4$ | $5.24 \times 10^4$ | $4.83 \times 10^4$ | $3.34 \times 10^4$ | $4.89 \times 10^4$ |
| **Amazon-ratings** | $3.23 \times 10^5$ | $5.23 \times 10^4$ | $1.13 \times 10^4$ | $1.12 \times 10^5$ | $5.23 \times 10^4$ | $1.03 \times 10^4$ | $1.90 \times 10^5$ | $5.23 \times 10^4$ | $1.07 \times 10^4$ |
| **Minesweeper** | NNE | NNE | NNE | NNE | NNE | NNE | NNE | NNE | NNE |

betweenness of different types of edges:

$$
\begin{aligned}
\mathsf{BetwIden}_q &:= \mathsf{Mean}(\mathsf{Between}(e), \forall e \in \mathcal{E}_q), \\
\mathsf{BetwAll} &:= \mathsf{Mean}(\mathsf{Between}(e), \forall e \in \mathcal{E}), \\
\mathsf{BetwIgno}_q &:= \mathsf{Mean}(\mathsf{Between}(e), \forall e \in \mathcal{E}, \mathsf{Curv}(e) \geq 0, \mathsf{JacoNorm}(e) \leq J_q).
\end{aligned}
\tag{5}
$$

BetwAll indicates the average betweenness of all edges, while BetwIden and BetwIgno represent the average betweenness of identified or ignored edges by curvature, respectively. See Table 3 for results.

> • **Observation 4**: In most cases, there is BetwAll > BetwIgno. This means that even within a cluster, there may be over-squashing edges, and discrete curvature ignores them systematically.
>
> • **Observation 5**: In most cases, there is BetwIden > BetwAll, which indicates that discrete curvature can only identify over-squashing edges that appear as "bridges" between clusters.

Previous studies have often, perhaps unconsciously, conflated terms such as "over-squashing" and "bottleneck" with "bridge edges connecting clusters" (a classical example can be found in Figure 1 of Topping et al. (2021)). However, our experimental results demonstrate that bottlenecks and over-squashing can also arise within clusters. The neglect of such edges by discrete curvature is precisely the key reason why curvature cannot serve as a necessary condition for detecting over-squashing. In this sense, curvature is not the gold standard for identifying over-squashing, but rather the gold standard for identifying bridge edges between clusters.

## 5 WEIGHTED AUGMENTED FORMAN-3 CURVATURE

In this section, we discuss how to leverage previous results to enhance existing discrete curvatures. According to the results in Table 2, we consider improving augmented Forman-3 curvature, which has the lowest time complexity and the best actual performance. We first provide an equivalent form for AF3 (Equation 6), where $u$ and $v$ are any pair of adjacent nodes in a graph, $\mathcal{N}(v) = \{i | i \sim v, i \in \mathcal{V}\}$ represents all first-order

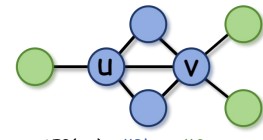

AF3(u,v) = #Blue - #Green

Figure 2: Calculation of AF3.

Table 4: The values of $\text{MOSR}_{10}$ and $\text{MOSR}_{25}$ across different GNNs and datasets when the discrete curvature si set to WAF3 and $f(x) \equiv 1/(1+x)$.

| | GCN | GAT | SAGE | | GCN | GAT | SAGE |
|---|---|---|---|---|---|---|---|
| **Cora** | .000/.014 | .157/.166 | .183/.183 | **Citeseer** | .020/.040 | .210/.216 | .279/.280 |
| **Pubmed** | .001/.002 | .013/.014 | .015/.015 | **Computers** | .002/.005 | .011/.011 | .011/.011 |
| **Photo** | .021/.024 | .027/.027 | .027/.027 | **CS** | .009/.034 | .075/.075 | .076/.076 |
| **Physics** | .020/.040 | .055/.058 | OOR | **WikiCS** | .140/.141 | .144/.144 | .144/.144 |
| **Cora_ML** | .025/.056 | .098/.103 | .103/.103 | **DBLP** | .020/.022 | .052/.055 | .057/.059 |
| **Cornell** | .000/.000 | .116/.122 | .138/.143 | **Texas** | .000/.000 | .134/.139 | .143/.149 |
| **Wisconsin** | .000/.000 | .108/.117 | .122/.134 | **Chameleon** | .065/.066 | .075/.076 | .079/.079 |
| **Squirrel** | .039/.039 | .041/.041 | .041/.041 | **Roman-empire** | .000/.001 | .351/.431 | .453/.453 |
| **Tolokers** | .002/.002 | OOR | .003/.003 | **Questions** | .003/.011 | .023/.024 | .025/.025 |
| **Amazon-ratings** | .159/.218 | .223/.231 | .245/.246 | **Minesweeper** | .191/.192 | .192/.192 | .192/.192 |
| | | | | **Average** | .036/.045 | .111/.118 | .123/.124 |

neighbors of $v$, and $\mathcal{B}(v) = \mathcal{N}(v) \cup \{v\}$ represents all nodes within a distance of 1 from $v$. The detailed derivation is provided in Appendix A.

$$\begin{aligned} \text{AF3}(u,v) &= 4 - d_u - d_v + 3\triangle(u,v) \\ &= \underbrace{|\mathcal{B}(u) \cap \mathcal{B}(v)|}_{\text{Number of nodes in triangles}} - \underbrace{(|\mathcal{N}(u)/\mathcal{B}(v)| + |\mathcal{N}(v)/\mathcal{B}(u)|)}_{\text{Number of nodes not in triangles}}. \end{aligned} \quad (6)$$

According to Equation 6, AF3 actually calculates the difference between the number of nodes that make up the triangle and the number of remaining first-order neighbors (Figure 2). However, as discussed in Chapter 3, triangle counting ignores the degree of nodes; however, high-degree nodes actually do little to enhance the information flow from the source node to the target node. Therefore, we propose weighted augmented Forman-3 curvature (WAF3). WAF3 weights each node's contribution to the curvature based on its degree by a function $f : \mathbb{R} \to \mathbb{R}$, which corrects for the influence of high-degree nodes. Obviously, AF3 is a special case of WAF3 when $f \equiv 1$.

$$\text{WAF3}_f(u,v) := \sum_{i \in \mathcal{B}(u) \cap \mathcal{B}(v)} f(d_i) - \left( \sum_{i \in \mathcal{N}(u)/\mathcal{B}(v)} f(d_i) + \sum_{i \in \mathcal{N}(v)/\mathcal{B}(u)} f(d_i) \right). \quad (7)$$

**Theorem 5** (WAF3 gets rid of counterexamples). *Consider an L-layer MPNN as in equation (1). Suppose $f(+\infty) = 0^+$. For the family of graphs $\mathscr{G} = \{\mathcal{G}_{n,m}^c\}_{n,m \in \mathbb{R}^+}$ with source-target pairs $(\text{s}, \text{t})$, There **dose not** exist a $c > 0$ such that for every $\mathcal{G} \in \mathscr{G}$, $\text{WAF3}(\text{s}, \text{t}) > c$.*

The above theorem clearly shows the difference between WAF3 and all other discrete curvatures, namely correcting the inappropriate contribution of high-degree nodes — only requiring the weight function $f$ to satisfy $f(+\infty) = 0^+$. Furthermore, we report the $\text{MOSR}_{10}$ and $\text{MOSR}_{10}$ values of AF3 in Table 4, where we set $f$ to be a GCN-style weighting function, i.e., $f = 1/(1+x)$, which satisfies the condition required in Theorem 5.

> • **Observation 6**: The $\text{MOSR}_{10}$ values of WAF3 range from 3.6% to 12.3%, while the $\text{MOSR}_{25}$ values range from 4.5% to 13.4%. This is a further reduction of more than 3% compared to the best-performing AF3 in Table 2.

## 6 ACCELERATING WAF3 VIA MINHASH

In terms of complexity, WAF3 has a time complexity of $\mathcal{O}(|\mathcal{E}| d_{\max})$, while WAF3 has $\mathcal{O}(|\mathcal{E}| d_{\max} \cdot \text{Complex}(f))$, in which the factor $d_{\max}$ comes from the intersection of sets $(\mathcal{B}(u) \cap \mathcal{B}(v))$. When $\text{Complex}(f) = \mathcal{O}(1)$ (such as the function $f = 1/(1+x)$ we use), the complexity of WAF3 is equivalent to that of AF3, which is the curvature with the lowest complexity at present (Table 1).

---

**Algorithm 1** Approximating WAF3

---

1: **Input:** Graph $\mathcal{G} = (\mathcal{V}, \mathcal{E})$, weighting function $f$, number of hashing $H$.
2: **for** $u \in \mathcal{V}$ **do**
3:  $u' = f(d_u)$;                        $\triangleright \Theta(|\mathcal{V}| \times \mathsf{Complex}(f))$
4: **end for**
5: **for** $u \in \mathcal{V}$ **do**
6:  $S_u = \{v'|v \in \mathcal{N}(u)\}$;
7:  $u'' = \mathsf{Sum}(S_u)$;                      $\triangleright \Theta(2|\mathcal{E}|)$
8: **end for**
9: **for** $(u, v) \in \mathcal{E}$ **do**
10:  $\mathsf{Jaccard}_f(\mathcal{N}(u), \mathcal{N}(v)) \approx \mathsf{Minhash}(S_u, S_v)$ (Ioffe, 2010);  $\triangleright \Theta(H|\mathcal{E}|)$
11:  Compute $\mathsf{WAF3}_f(u, v)$ via Theorem 6;            $\triangleright \Theta(|\mathcal{E}|)$
12: **end for**

---

However, this is still unacceptable in large graphs, since $d_{\max}$ usually also grows with the number of nodes in graphs. This dilemma has largely hindered the promotion of curvature tools to graph learning. To solve this problem, we first prove that an equivalent form of WAF3 is as follows:

**Theorem 6** (Equivalent form of WAF3 based on Jaccard similarity). *Let weighted Jaccard similarity* $\mathsf{Jaccard}_f(\mathcal{N}(u), \mathcal{N}(v)) := \frac{\sum_{i \in \mathcal{N}(u) \cap \mathcal{N}(v)} f(d_i)}{\sum_{i \in \mathcal{N}(u) \cup \mathcal{N}(v)} f(d_i)}$, *then the following equation holds:*

$$\mathsf{WAF3}_f(u, v) = 2f(u) + 2f(v)$$
$$+ \left(2 - \frac{3}{1 + \mathsf{Jaccard}_f(\mathcal{N}(u), \mathcal{N}(v))}\right) \left(\sum_{i \in \mathcal{N}(u)} f(d_i) + \sum_{i \in \mathcal{N}(v)} f(d_i)\right).$$

Theorem 6 converts the time-consuming intersection operation into a weighted Jaccard similarity operation. Fortunately, the acceleration algorithm of the latter has been widely studied (Wu et al., 2020). In particular, a class of algorithms called weighted Minhash (Manasse, 2010; Ioffe, 2010; Wu et al., 2016; 2017; 2018) can reduce the complexity of computing Jaccard similarity to a constant ($\mathcal{O}(H)$) by sampling $H$ hash functions, where the larger $H$ is, the smaller the approximation error is. At this point, the overall complexity of WAF3 will be further advanced to $\mathcal{O}(H|\mathcal{E}|)$, reaching its theoretical lower bound (because it has constant complexity for each edge).

Two experiments are conducted to evaluate the practicality of Algorithm 1. First, in order to verify the computational efficiency, we randomly generated Erdős–Rényi random graphs (ERDdS & R&wi, 1959) with the number of nodes being $\{10^4, 2 \times 10^4, 3 \times 10^4, 4 \times 10^4, 5 \times 10^4, 10^5\}$, where the connection probability $p$ between any two nodes was set to 0.0005, which is comparable to the sparsity level of most commonly used graph datasets. Under the constraint of a maximum GPU memory of 24GB, the running times of different curvature computations are reported in Figure 3.

Secondly, we evaluate whether the approximation error is within an acceptable range. It is worth noting that, for most curvature-based graph learning

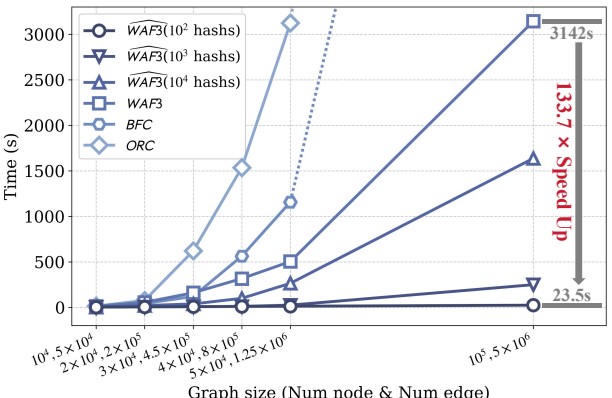

Figure 3: Computation time of different discrete curvatures with $p = 0.0005$, $10^4$–$10^5$ nodes, and 24 GB GPU limit. Here, $\widehat{\mathsf{WAF3}}$ denotes the WAF3 approximation with different number of hashing (Algorithm 1).

methods, the relative ordering of curvature values is often more critical than their absolute magnitudes (for example, in rewiring-based approaches (Nguyen et al., 2023; Giraldo et al., 2023; Fesser &

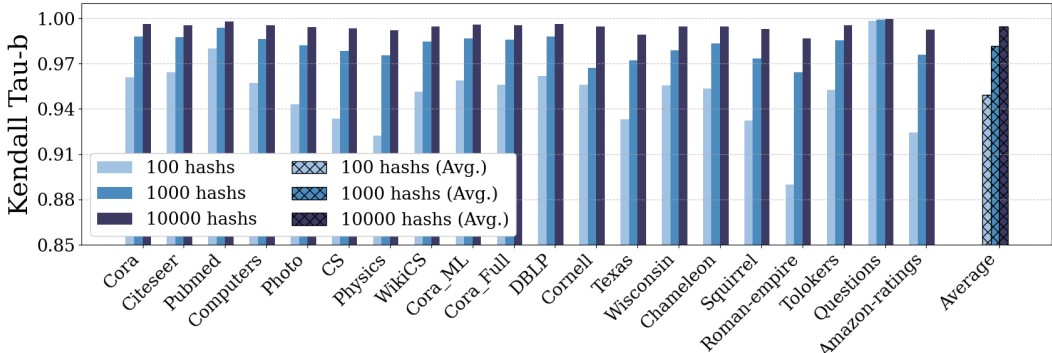

Figure 4: The Kendall Tau-b similarity between WAF3 and $\widehat{\text{WAF3}}$ values calculated for different datasets when $H$ is 100, 1000, and 10000. The average value is also reported.

Weber, 2024), typically only the edges with the smallest curvature are considered). Based on this observation, we report in Figure 4 the Kendall Tau-b similarity (Kendall, 1938) between WAF3 and its approximated values. This rank-based metric measures the consistency between two sequences, with higher values indicating greater agreement in their orderings.

> • **Observation 7**: When the random graph scales to $10^5$ nodes and $5 \times 10^6$ edges, ORC ($\mathcal{O}(|\mathcal{E}|d_{\max}^3)$) and BFC ($\mathcal{O}(|\mathcal{E}|d_{\max}^2)$) can no longer complete the computation within a tolerable time. WAF3 $\mathcal{O}(|\mathcal{E}|d_{\max})$ also requires over 3,000 seconds. However, the Minhash-based approximation algorithm has a significant acceleration effect, especially when $H = 100$, with a speedup of 133.7 times.
>
> • **Observation 8**: Even when $H = 100$, the average Kendall tau-b similarity across all datasets is approximately 95%. This means that only $(100\% - 95\%)/2 = 2.5\%$ of edge pairs are misordered. When $H = 1000$, the similarity rises to over 98%. On this basis, the similarity improvement for $H = 10,000$ is less than 1%, a significant marginal benefit.

According to Figure 3, for large-scale graphs, even the exact computation of the least complex curvature becomes prohibitively expensive. However, our proposed algorithm facilitates the emergence of a new paradigm for curvature-based learning: first, a small subset of candidate edges is identified through an approximation algorithm, and then the final set of highly negatively curved edges is determined via exact computation.

We provide more experiments and observations in Appendix, including the impact of model training and the value of $q$ on MOSR (Appendix D.1); insights into designing the weighting function $f$ (Appendix D.2); and the effect of WAF3 in actual curvature garph learning (Appendix D.3).

## 7 CONCLUSION

This work revisits the belief that discrete curvature reliably captures over-squashing in graph neural networks. We prove through constructive counterexamples that even highly positive-curvature edges can suffer from severe squashing, showing that curvature is not a necessary condition. To quantify this gap, we introduce MOSR, a metric that measures the proportion of over-squashed edges missed by curvature-based criteria, and we find that common curvatures such as Ollivier–Ricci may overlook more than 30%. We further present WAF3, a weighted refinement of Forman-3 curvature, which addresses the theoretical limitations of existing definitions. By reformulating it as a weighted Jaccard similarity and applying weighted MinHash, we achieve practical scalability with over two orders of magnitude speedup.

Overall, this work both challenges an implicit assumption in the community and provides a feasible alternative. It encourages more principled ways to characterize over-squashing and supports the design of graph learning methods that move beyond curvature as a universal surrogate.

## 8 REPRODUCIBILITY STATEMENT

We provide detailed proofs and derivations of all theoretical results presented in the main text in the appendix. We also provide an anonymous link to the code of this project and state all necessary implementation details in the appendix. We provide a statement of use for the LLMs at the end of the appendix.

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

# A  PROOFS & DERIVATIONS

## A.1  PROOF OF LEMMA 2

**Lemma 7.** *Let* $\mathbf{A}$ *be the adjacency matrix of an undirected, unweighted, simple graph, and* $\mathbf{D}$ *the degree matirx. For any two adjacent points* $i$ *and* $j$ *in the graph, for any* $L \in \mathbb{Z}^+$ *we have:*

$$\left((\mathbf{D} + \mathbf{I})^{-1/2}(\mathbf{A} + \mathbf{I})(\mathbf{D} + \mathbf{I})^{-1/2}\right)_{ij}^L \geq \left(\frac{1}{d_i + 1} + \frac{1}{d_j + 1}\right)^{L-1} \frac{1}{\sqrt{(d_i + 1)(d_j + 1)}}.$$

*The equal sign is obtained iff* $d_i = d_j = 1$.

*Proof.* Define $\mathbf{M} = (\mathbf{D} + \mathbf{I})^{-1/2}(\mathbf{A} + \mathbf{I})(\mathbf{D} + \mathbf{I})^{-1/2}$. The element of $\mathbf{M}$ is given by:

$$\mathbf{M}_{ij} = \begin{cases} \dfrac{1}{d_i + 1} & \text{for the diagonal,} \\ \dfrac{1}{\sqrt{(d_i + 1)(d_j + 1)}} & \text{if } i \sim j, \\ 0 & \text{otherwise.} \end{cases}$$

The $(i, j)$-th entry of $\mathbf{M}^L$, denoted $\mathbf{M}_{ij}^L$, represents the sum of the weights of all walks of length $L$ from node $i$ to node $j$. The weight of a walk is the product of the M-weights of the edges traversed at each step. To bound $\mathbf{M}_{ij}^L$, we consider the subgraph restricted to nodes $i$ and $j$. Since $i$ and $j$ are adjacent, this subgraph includes the edge $(i, j)$, along with self-loops at both nodes. The matrix $\mathbf{M}$ restricted to these two nodes is:

$$\mathbf{P} = \begin{bmatrix} \mathbf{M}_{ii} & \mathbf{M}_{ij} \\ \mathbf{M}_{ji} & \mathbf{M}_{jj} \end{bmatrix} = \begin{bmatrix} \frac{1}{d_i+1} & \frac{1}{\sqrt{(d_i+1)(d_j+1)}} \\ \frac{1}{\sqrt{(d_j+1)(d_j+1)}} & \frac{1}{d_j+1} \end{bmatrix}.$$

We now compute the $(1, 2)$-th entry of $\mathbf{P}^L$, which corresponds to walks from $i$ (index 1) to $j$ (index 2) within this subgraph. Note that $\mathbf{P}$ can be expressed as the outer product $\mathbf{P} = \mathbf{v}\mathbf{v}^T$, where:

$$\mathbf{v} = \begin{bmatrix} \frac{1}{\sqrt{d_i+1}} & \frac{1}{\sqrt{d_j+1}} \end{bmatrix}^T.$$

Thus for any integer $L \geq 1$:

$$\mathbf{P}^L = (\mathbf{v}\mathbf{v}^T)^L = (\mathbf{v}^T\mathbf{v})^{L-1}(\mathbf{v}\mathbf{v}^T) = \left(\frac{1}{d_i + 1} + \frac{1}{d_j + 1}\right)^{L-1} \mathbf{P}.$$

Specifically, we have:

$$\mathbf{P}_{12}^L = \left(\frac{1}{d_i + 1} + \frac{1}{d_j + 1}\right)^{L-1} \frac{1}{\sqrt{(d_i + 1)(d_j + 1)}}.$$

In the full graph, $\mathbf{M}_{ij}^L$ includes all walks of length $L$ from $i$ to $j$, not only those confined to $\{i, j\}$. The walks restricted to $\{i, j\}$ form a subset of these walks, and their total weight is exactly $\mathbf{P}_{12}^L$. Since all entries of $\mathbf{M}$ are non-negative ($\mathbf{M}_{uv} \geq 0$ for all $u, v$), the weight of every walk is non-negative. Therefore, the sum over all walks is at least the sum over the subset of walks confined to $\{i, j\}$:

$$\mathbf{M}_{ij}^L \geq \mathbf{P}_{12}^L.$$

This establishes the desired inequality. The equal sign holds iff $\mathbf{P} \equiv \mathbf{M}$, which means $d_i = d_j = 1$. $\qquad\square$

Before we prove Lemma 2, we first further explain Assumption 1. When we compute the $\partial \mathbf{h}_\mathsf{t}^{(L)} / \partial \mathbf{h}_\mathsf{s}^{(0)}$, we obtain a sum of different terms over all possible paths from s to t of length $L$. In this case, the derivative of ReLU acts as a Bernoulli variable evaluated along all these possible paths. So follow the very same argument in Di Giovanni et al. (2023a); Kawaguchi (2016); Xu et al. (2018), we can get:

$$\mathbb{E}\left[\frac{\partial \mathbf{h}_\mathsf{t}^{(L)}}{\partial \mathbf{h}_\mathsf{s}^{(0)}}\right] = \rho \prod_{l=0}^{L-1} \mathbf{W}^{(l)} \left((\mathbf{D} + \mathbf{I})^{-1/2}(\mathbf{A} + \mathbf{I})(\mathbf{D} + \mathbf{I})^{-1/2}\right)_{\mathsf{s},\mathsf{t}}^L$$

Which the expectation means that we are taking the average over such Bernoulli variables. Then we can take the norm (in expectation) and leverage Lemma 5:

$$\left\| \frac{\partial \mathbf{h}_t^{(L)}}{\partial \mathbf{h}_s^{(0)}} \right\| \geq \rho \left\| \prod_{l=0}^{L-1} \mathbf{W}^{(l)} \right\| \left( \frac{1}{a+1} + \frac{1}{b+1} \right)^{L-1} \frac{1}{\sqrt{(a+1)(b+1)}}$$

Which $a := d_s$ and $b := d_t$. Note that the right side of the inequality has nothing to do with the graph structure. And by Lemma 5, this bound is tight. So the proof is complete.

## A.2 PROOF OF THEOREM 4

**Lemma 8.** *Consider a family of $n \times n$ matrices $\mathbf{A}^{(\alpha)} = \{a_{ij}^{(\alpha)}\}$ parameterized by $\alpha > 0$, satisfying:*

- **Diagonal invariance**: *For all $1 \leq i \leq n$, the diagonal elements $a_{ii}^{(\alpha)}$ are constant (independent of $\alpha$).*
- **Upper-left block invariance**: *There exists a fixed integer $m$ $(1 \leq m \leq n)$ such that the submatrix $\mathbf{A}_{\mathsf{sub}} = (a_{ij}^{(\alpha)})_{1 \leq i,j \leq m}$ is constant (independent of $\alpha$).*
- **Uniform decay outside block**: *All other elements are bounded by $\alpha$, i.e.,*

$$|a_{ij}^{(\alpha)}| \leq \alpha \quad \text{for all} \quad (i > m \text{ or } j > m) \text{ and } i \neq j.$$

*Let $\mathbf{A}^{k,(\alpha)}$ denote the $k$-th power of $\mathbf{A}^{(\alpha)}$, and $\mathbf{A}_{\mathsf{sub}}^k$ the $k$-th power of the fixed submatrix $\mathbf{A}_{\mathsf{sub}}$. Then for any fixed integer $k \geq 1$ and all $1 \leq i, j \leq m$:*

$$\lim_{\alpha \to 0} \left| (\mathbf{A}^{k,(\alpha)})_{ij} - (\mathbf{A}_{\mathsf{sub}}^k)_{ij} \right| = 0$$

*with convergence rate $\mathcal{O}(\alpha^2)$.*

*Proof.* Fix $k \geq 1$ and $i, j \in \{1, \cdots, m\}$. We interpret $\mathbf{A}^{(\alpha)}$ as the adjacency matrix of a weighted directed graph on $n$ nodes, where $a_{ij}^{(\alpha)}$ is the edge weight from node $i$ to $j$. The $(i, j)$-entry of $\mathbf{A}^{(\alpha),k}$ equals the sum of weights of all paths of length $k$ from $i$ to $j$, with path weight defined as the product of edge weights. Similarly, $(\mathbf{A}_{\mathsf{sub}}^k)_{ij}$ sums weights of paths confined to the subgraph of nodes $\{1, \cdots, m\}$.

Let $\mathcal{P}_{\mathsf{sub}}$ be the set of paths from $i$ to $j$ of length $k$ within $\{1, \cdots, m\}$, and $\mathcal{P}_{\mathsf{else}}$ the set that visits at least one node in $\{m+1, \cdots, n\}$. Then:

$$(\mathbf{A}^{k,(\alpha)})_{ij} = \sum_{p \in \mathcal{P}_{\mathsf{sub}}} w(p) + \sum_{p \in \mathcal{P}_{\mathsf{else}}} w(p), \quad (\mathbf{A}_{\mathsf{sub}}^k)_{ij} = \sum_{p \in \mathcal{P}_{\mathsf{sub}}} w(p),$$

so:

$$\left| (\mathbf{A}^{k,(\alpha)})_{ij} - (\mathbf{A}_{\mathsf{sub}}^k)_{ij} \right| = \left| \sum_{p \in \mathcal{P}_{\mathsf{else}}} w(p) \right| = \sum_{p \in \mathcal{P}_{\mathsf{else}}} |w(p)|.$$

Where $w(p)$ denotes the weight of the path $p$, which is always nonnegative. for each path $p \in \mathcal{P}_{\mathsf{else}}$, it contains at least one edge that move from node set $\{1, \cdots, m\}$ to node set $\{m+1, \cdots, n\}$, and also contains at least one edge that move from node set $\{m+1, \cdots, n\}$ to node set $\{1, \cdots, m\}$. Thus:

$$w(p) \leq \alpha^2 \cdot \left( \max \left\{ \alpha, \max_i \{a_{ii}^{(\alpha)}\}, \max_{1 \leq i,j \leq m} \{a_{ik}^{(\alpha)}\} \right\} \right)^{k-2}.$$

When $\alpha \leq \max_i \{a_{ii}^{(\alpha)}\}$ and $\alpha \leq \max_{1 \leq i,j \leq m} \{a_{ik}^{(\alpha)}\}$, the above formula is simplified to:

$$w(p) \leq \alpha^2 \cdot \left( \max \left\{ \max_i \{a_{ii}^{(\alpha)}\}, \max_{1 \leq i,j \leq m} \{a_{ik}^{(\alpha)}\} \right\} \right)^{k-2} = \alpha^2 C_1^{k-2}.$$

Because $|\mathcal{P}_{\mathsf{path}}| + |\mathcal{P}_{\mathsf{else}}| = n^{k-1}$ and $|\mathcal{P}_{\mathsf{path}}| = m^{k-1}$, so

$$\left| (\mathbf{A}^{k,(\alpha)})_{ij} - (\mathbf{A}_{\mathsf{sub}}^k)_{ij} \right| \leq \alpha^2 C_1^{k-2} (n^{k-1} - m^{k-1}) = \alpha^2 C_2$$

which implies $\lim_{\alpha \to 0} \left| (\mathbf{A}^{k,(\alpha)})_{ij} - (\mathbf{A}_{\mathsf{sub}}^k)_{ij} \right| = 0$ with convergence rate $\mathcal{O}(\alpha^2)$. $\qquad\square$

We first prove the second property in Theorem 3, namely the convergence of $\|\partial \mathbf{h}_t^{(L)}/\partial \mathbf{h}_s^{(0)}\|$. Let $\mathbf{A}^c$ the adjacent matrix of $\mathcal{G}_{n,m}^c$, and $\mathbf{D}^c$ the degree matrix. Similar to the proof of Lemma 2, we have:

$$\left\| \frac{\partial \mathbf{h}_t^{(L)}}{\partial \mathbf{h}_s^{(0)}} \right\| = \rho \left\| \prod_{l=0}^{L-1} \mathbf{W}^{(l)} \right\| \left( (\mathbf{D}^c + \mathbf{I})^{-1/2} (\mathbf{A}^c + \mathbf{I})(\mathbf{D}^c + \mathbf{I})^{-1/2} \right)_{s,t}^L$$

Let $\mathbf{M}^c := (\mathbf{D}^c + \mathbf{I})^{-1/2}(\mathbf{A}^c + \mathbf{I})(\mathbf{D}^c + \mathbf{I})^{-1/2}$, we have:

$$\mathbf{M}_{ij}^c = \begin{cases} \dfrac{1}{n+2} & \text{if } i \in \{s, t\} \text{ and } j \in \{s, t\}, \\[2mm] \dfrac{1}{m+3} & \text{if } i = j \in \mathcal{N}_1, \\[2mm] \dfrac{1}{2} & \text{if } i = j \in \mathcal{N}_2, \\[2mm] \dfrac{1}{\sqrt{(n+2)(m+3)}} & \text{if } (i \in \{s, t\} \text{ and } j \in \mathcal{N}_1) \text{ or } (i \in \mathcal{N}_1 \text{ and } j \in \{s, t\}), \\[2mm] \dfrac{1}{\sqrt{2(m+3)}} & \text{if } (i \in \mathcal{N}_1 \text{ and } j \in \mathcal{N}_2) \text{ or } (i \in \mathcal{N}_2 \text{ and } j \in \mathcal{N}_1), \\[2mm] 0 & \text{otherwise.} \end{cases}$$

Since

$$\max\left( 0, \frac{1}{\sqrt{(n+2)(m+3)}}, \frac{1}{\sqrt{2(m+3)}} \right) = \frac{1}{\sqrt{2(m+3)}},$$

According to Lemma 6, we have:

$$(\mathbf{M}^{c,L})_{s,t} \to \left( \begin{bmatrix} 1/(n+2) & 1/(n+2) \\ 1/(n+2) & 1/(n+2) \end{bmatrix}^L \right)_{1,2} = \frac{2^{L-1}}{(n+2)^L} = \phi_L(n+1, n+1).$$

with convergence rate $\mathcal{O}(1/(2(m+3))) = \mathcal{O}(m^{-1})$ when $m \to +\infty$.

Then we prove separately that for all $n, m \in \mathbb{R}^+$, the seven discrete curvatures (except link resistance curvature) between nodes s and t are always positive in graph $\mathcal{G}_{n,m}^c$.

**Augmented Forman-3 curvature & Augmented Forman-4 curvature** Note that edge $(s, t)$ is not contained in any 4-cycle in $\mathcal{G}_{n,m}^c$, thus $\square(s, t) = 0$. Thus for all $n, m \in \mathbb{R}^+$, we have:

$$\mathsf{AF3}(s, t) = \mathsf{AF4}(s, t) = 4 - n - n + 3n = 4 + n \geq 5.$$

**Jost-Liu forman curvature** Note that $\triangle(s, t) = n$ and $d_s \wedge d_t = d_s \vee d_t = n+1$. Thus for all $\mathcal{G}_{n,m}^c$:

$$\begin{aligned} \mathsf{JLF}(s, t) &= -\left(1 - \frac{1}{d_s} - \frac{1}{d_t} - \frac{\triangle(s, t)}{d_s \vee d_t}\right)_+ - \left(1 - \frac{1}{d_s} - \frac{1}{d_t} - \frac{\triangle(s, t)}{d_s \wedge d_t}\right)_+ + \frac{\triangle(s, t)}{d_s \vee d_t} \\ &= -\left(1 - \frac{1}{n+1} - \frac{1}{n+1} - \frac{n}{n+1}\right)_+ - \left(1 - \frac{1}{n+1} - \frac{1}{n+1} - \frac{n}{n+1}\right)_+ + \frac{n}{n+1} \\ &= \frac{n}{n+1} \geq \frac{1}{2}. \end{aligned}$$

**Balance forman curvature with/without 4-cycle** For all $n, m \in \mathbb{R}^+$:

$$\begin{aligned} \mathsf{BFw/o4}(s, t) &= \frac{2}{d_s} + \frac{2}{d_t} - 2 + 2\frac{\triangle(s, t)}{d_s \vee d_t} + \frac{\triangle(s, t)}{d_s \wedge d_t} \\ &= \frac{2}{n+1} + \frac{2}{n+1} - 2 + 2\frac{n}{n+1} + \frac{n}{n+1} \\ &= \frac{n+2}{n+1} > 1. \end{aligned}$$

Note that $\frac{\#_\square^s(s,t) + \#_\square^t(s,t)}{\gamma(s,t)(d_s \vee d_t)} \geq 0$, Thus:

$$\mathsf{BF}(s, t) \geq \mathsf{BFw/o4}(s, t) > 1.$$

**Ollivier Ricci curvature** Note that:

$$\mu_{\mathsf{s}}^{\alpha}(v) = \begin{cases} \alpha & v = \mathsf{s}, \\ \dfrac{1-\alpha}{n+1} & v \in \mathcal{N}_1 \cup \{\mathsf{t}\}, \\ 0 & \text{else.} \end{cases}$$

and

$$\mu_{\mathsf{t}}^{\alpha}(v) = \begin{cases} \alpha & v = \mathsf{t}, \\ \dfrac{1-\alpha}{n+1} & v \in \mathcal{N}_1 \cup \{\mathsf{s}\}, \\ 0 & \text{else.} \end{cases}$$

To calculate $W_1(\mu_{\mathsf{s}}^{\alpha}, \mu_{\mathsf{t}}^{\alpha})$, when $\alpha > \frac{1-\alpha}{n+1}$, we need to move $\alpha - \frac{1-\alpha}{n+1}$ probability mass directly from s to t; when $\alpha < \frac{1-\alpha}{n+1}$, we need to move $\frac{1-\alpha}{n+1} - \alpha$ probability mass directly from t to s. So $W_1(\mu_{\mathsf{s}}^{\alpha}, \mu_{\mathsf{t}}^{\alpha}) = |\alpha - \frac{1-\alpha}{n+1}|$ and $\alpha\mathsf{OR}(\mathsf{s}, \mathsf{t}) = 1 - |\alpha - \frac{1-\alpha}{n+1}|$ and thus $\mathsf{OR}(\mathsf{s}, \mathsf{t}) = 1 - \frac{1}{n+1} \geq \frac{1}{2}$ when we take $\alpha = 0$.

**Lin-Lu-Yau curvature** By using the definition:

$$\mathsf{LLY}(\mathsf{s}, \mathsf{t}) = \lim_{\alpha \to 1^-} \frac{\alpha\mathsf{OR}(\mathsf{s}, \mathsf{t})}{1 - \alpha} = \lim_{\alpha \to 1^-} \frac{1 - |\alpha - \frac{1-\alpha}{n+1}|}{1 - \alpha}$$

Define $g(\alpha) = \alpha - \frac{1-\alpha}{n+1}$. For $\alpha$ sufficiently close to 1, specifically when $\alpha > \frac{1}{n+2}$ (which holds for $\alpha$ in a left neighborhood of 1 since $\frac{1}{n+2} < 1$ for all positive integers $n$), we have $g(\alpha) \geq 0$. Thus, $|g(\alpha)| = g(\alpha)$ in this region. The numerator simplifies as follows:

$$1 - |g(\alpha)| = 1 - g(\alpha) = 1 - \left( \alpha - \frac{1-\alpha}{n+1} \right) = (1 - \alpha)\frac{n+2}{n+1}.$$

Therefore:

$$\lim_{\alpha \to 1^-} \frac{1 - |g(\alpha)|}{1 - \alpha} = \lim_{\alpha \to 1^-} \frac{(1-\alpha)\frac{n+2}{n+1}}{1 - \alpha} = \lim_{\alpha \to 1^-} \frac{n+2}{n+1} = \frac{n+2}{n+1} > 1.$$

Which means $\mathsf{LLY}(\mathsf{s}, \mathsf{t}) > 1$.

**Link resistance curvature** Finally, we prove that in link resistance curvature, for any fixed $n^*$, there exists an $m^*$ such that for all $m > m^*$, and node-pair $(\mathsf{s}, \mathsf{t})$ in $\mathcal{G}_{n^*, m}^{\mathsf{c}}$, $\mathsf{LR}(\mathsf{s}, \mathsf{t}) > 0$.

To calculate the link resistance curvature, we first need to calculate the equivalent resistance of each edge in $\mathcal{G}_{n,m}^{\mathsf{c}}$. Looking back at the definition of $\mathcal{G}_{n,m}^{\mathsf{c}}$, there are three equivalent types of edges. Let's assume that the edges in $\{(\mathsf{s}, \mathsf{t})\}$, $\{(\mathsf{s}, u_i), (\mathsf{t}, u_i)\}_{1 \leq i \leq n}$, and $\{(u_i, v_{ij})\}_{1 \leq i \leq n, 1 \leq j \leq m}$ have resistance values of $r_1$, $r_2$, and $r_3$ respectively. And their equivalent resistances are recorded as $w_1$, $w_2$, and $w_3$ respectively.

Calculate $w_1$: Note that there are $n + 1$ resistors in parallel between s and t, $n$ of which have a resistance of $2r_2$ and one has a resistance of $r_1$, so:

$$w_1 = \left( \frac{1}{r_1} + \frac{n}{2r_2} \right)^{-1}.$$

Calculate $w_2$: Since the equivalent resistance of all edges in $\{(\mathsf{s}, u_i), (\mathsf{t}, u_i)\}_{1 \leq i \leq n}$ is equal, let's take the calculation of the equivalent resistance between s and $u_1$ as an example. According to the topology of the graph, the calculation is divided into three steps. First, we connect $n$ resistors in parallel: $\mathsf{s} - \mathsf{t}, \mathsf{s} - u_2 - \mathsf{t}, \mathsf{s} - u_3 - \mathsf{t}, \cdots, \mathsf{s} - u_n - \mathsf{t}$, where the resistance of $\mathsf{s} - \mathsf{t}$ is $r_1$ and the resistance of the remaining $n - 1$ resistors is $2r_2$. Then, we connect the resulting resistor in series with $\mathsf{t} - u_1$ (with a resistance of $r_2$). Finally, we connect the resulting resistor in parallel with $\mathsf{s} - u_1$ (with a resistance of $r_2$). Therefore, $w_2$ is calculated as:

$$w_2 = \left[ \frac{1}{r_2 + \left( \frac{1}{r_1} + \frac{n-1}{2r_2} \right)^{-1}} + \frac{1}{r_2} \right]^{-1}.$$

Calculate $w_3$: Since any $v_{ij}$ is only connected to $u_i$, the equivalent resistance between $v_{ij}$ and $u_i$ is not affected by any other resistance, so:

$$w_3 = r_3.$$

From the definition,

$$\mathsf{LR(s,t)} = \frac{2(1 - nw_2)}{w_1}.$$

since $w_1 > 0$, making $\mathsf{LR(s,t)} > 0$ is equivalent to making $1 - nw_2 > 0$ true, which is further equivalent to ensure:

$$\left[ r_2 + \left( \frac{1}{r_1} + \frac{n-1}{2r_2} \right)^{-1} \right]^{-1} + \frac{1}{r_2} > n.$$

Simplify the left side of the above inequality:

$$\text{left} = \left[ r_2 + \left( \frac{2r_2 + (n-1)r_1}{2r_1 r_2} \right)^{-1} \right]^{-1} + \frac{1}{r_2} = \left[ r_2 + \frac{2r_1 r_2}{2r_2 + (n-1)r_1} \right]^{-1} + \frac{1}{r_2}$$

$$= \left[ \frac{2r_2^2 + (n+1)r_1 r_2}{2r_2 + (n-1)r_1} \right]^{-1} + \frac{1}{r_2} = \frac{2r_2 + (n-1)r_1}{2r_2^2 + (n+1)r_1 r_2} + \frac{1}{r_2} = \frac{2r_2 + 2nr_1}{2r_2^2 + (n+1)r_1 r_2}.$$

Considering that in the symmetric normalized Laplacian, the value of any edge $(u, v)$ is $\frac{1}{\sqrt{(d_u+1)(d_v+1)}}$, so $r_1 = \frac{1}{n+1}$ and $r_2 = \frac{1}{\sqrt{(n+1)(m+2)}}$. Thus $(n+1)r_1 = 1$. We can further deduce the equivalent conditions as follows:

$$\frac{2r_2 + 2nr_1}{2r_2^2 + (n+1)r_1 r_2} > n \Rightarrow \frac{2r_2 + 2nr_1}{2r_2^2 + r_2} > n \Rightarrow (2 - n)r_2 + 2nr_1 - 2nr_2^2 > 0.$$

We further narrow the left side of the above inequality to make its validity condition more stringent:

$$(2 - n)r_2 + 2nr_1 - 2nr_2^2 > -nr_2 + 2nr_1 - 2nr_2^2 = n(2r_1 - r_2 - 2r_2^2)$$

$$= n \left( \frac{2}{n+1} - \frac{1}{\sqrt{(n+1)(m+2)}} - \frac{2}{(n+1)(m+2)} \right) := n \times f(n, m).$$

Obviously, $\partial f(n, m) / \partial m > 0$, and $f(n, +\infty) = \frac{2}{n+1} > 0$, so for any fixed $n^*$, there must exist a $m^*$ such that for any $m > m^*$, $f(n^*, m)$ approaches $\frac{2}{n+1}$ arbitrarily, thus the original proposition is proved.

### A.3 DERIVATION OF EQUATION 6

$$4 - d_i - d_j + 3\triangle_{ij}$$
$$= 4 - |\mathcal{N}(u)| - |\mathcal{N}(v)| + 3(|\mathcal{N}(u) \cap \mathcal{N}(v)|)$$
$$= 4 - (|\mathcal{N}(u)| - |\mathcal{N}(u) \cap \mathcal{N}(v)|) - (|\mathcal{N}(v)| - |\mathcal{N}(u) \cap \mathcal{N}(v)|) + |\mathcal{N}(u) \cap \mathcal{N}(v)|$$
$$= 4 - |\mathcal{N}(u)/\mathcal{N}(v)| - |\mathcal{N}(v)/\mathcal{N}(u)| + |\mathcal{N}(u) \cap \mathcal{N}(v)|$$
$$= (2 + |\mathcal{N}(u) \cap \mathcal{N}(v)|) - (|\mathcal{N}(u)/\mathcal{N}(v)| - 1) - (|\mathcal{N}(v)/\mathcal{N}(u)| - 1)$$
$$= |\mathcal{B}(u) \cap \mathcal{B}(v)| - |\mathcal{N}(u)/\mathcal{B}(v)| - |\mathcal{N}(v)/\mathcal{B}(u)|$$

### A.4 PROOF OF THEOREM 5

By definition, we know that for any $\mathcal{G}_{n,m}^c$, we have:

$$\mathsf{WAF3(s,t)} = 2f(n+1) + nf(m+2)$$

The original proposition is equivalent to proving that for any $\epsilon > 0$, there exists a set of positive integers $n^*$ and $m^*$ such that $2f(n+1) + nf(m+2) < \epsilon$. Since $f(+\infty) = 0^+$, for any $\epsilon > 0$, there exists $N_1 > 0$ such that when $x > N_1$, we have $f(x) < \frac{\epsilon}{4}$. Choose a positive integer $n^*$ such that $n^* + 1 > N_1$. Then, $f(n^* + 1) < \frac{\epsilon}{4}$, and thus

$$2f(n^* + 1) < \frac{\epsilon}{2}.$$

For a fixed $n^*$, since $f(x) \to 0$ as $x \to \infty$, there exists $N_2 > 0$ such that when $x > N_2$, we have $f(x) < \frac{\epsilon}{2n^*}$. Choose a positive integer $m^*$ such that $m^* + 2 > N_2$. Then, $f(m^* + 2) < \frac{\epsilon}{2n^*}$, and thus

$$n^* f(m^* + 2) < \frac{\epsilon}{2}.$$

Therefore,

$$2f(n^* + 1) + n^* f(m^* + 2) < \frac{\epsilon}{2} + \frac{\epsilon}{2} = \epsilon.$$

### A.5 Proof of Theorem 6

Notice that:

$$\mathsf{Jaccard}_f(\mathcal{N}(u), \mathcal{N}(v)) = \frac{\sum_{i \in \mathcal{N}(u) \cap \mathcal{N}(v)} f(d_i)}{\sum_{i \in \mathcal{N}(u) \cup \mathcal{N}(v)} f(d_i)}$$

$$\Longleftrightarrow \mathsf{Jaccard}_f(\mathcal{N}(u), \mathcal{N}(v)) = \frac{\sum_{i \in \mathcal{N}(u) \cap \mathcal{N}(v)} f(d_i)}{\sum_{i \in \mathcal{N}(u)} f(d_i) + \sum_{i \in \mathcal{N}(v)} f(d_i) - \sum_{i \in \mathcal{N}(u) \cap \mathcal{N}(v)} f(d_i)}$$

$$\Longleftrightarrow \frac{1}{\mathsf{Jaccard}_f(\mathcal{N}(u), \mathcal{N}(v))} = \frac{\sum_{i \in \mathcal{N}(u)} f(d_i) + \sum_{i \in \mathcal{N}(v)} f(d_i)}{\sum_{i \in \mathcal{N}(u) \cap \mathcal{N}(v)} f(d_i)} - 1$$

$$\Longleftrightarrow \sum_{i \in \mathcal{N}(u) \cap \mathcal{N}(v)} f(d_i) = \frac{\sum_{i \in \mathcal{N}(u)} f(d_i) + \sum_{i \in \mathcal{N}(v)} f(d_i)}{\mathsf{Jaccard}_f^{-1}(\mathcal{N}(u), \mathcal{N}(v)) + 1}$$

So:

$$\mathsf{WAF3}_f(u, v)$$

$$= \sum_{i \in \mathcal{B}(u) \cap \mathcal{B}(v)} f(d_i) - \sum_{i \in \mathcal{N}(u)/\mathcal{B}(v)} f(d_i) - \sum_{i \in \mathcal{N}(v)/\mathcal{B}(u)} f(d_i)$$

$$= \left( f(d_u) + f(d_v) + \sum_{i \in \mathcal{N}(u) \cap \mathcal{N}(v)} f(d_i) \right) - \left( \sum_{i \in \mathcal{N}(u)/\mathcal{N}(v)} f(d_i) - f(d_v) \right) - \left( \sum_{i \in \mathcal{N}(v)/\mathcal{N}(u)} f(d_i) - f(d_u) \right)$$

$$= 2f(d_u) + 2f(d_v) + \sum_{i \in \mathcal{N}(u) \cap \mathcal{N}(v)} f(d_i) - \sum_{i \in \mathcal{N}(u)/\mathcal{N}(v)} f(d_i) - \sum_{i \in \mathcal{N}(v)/\mathcal{N}(u)} f(d_i)$$

$$= 2f(d_u) + 2f(d_v) + \sum_{i \in \mathcal{N}(u) \cap \mathcal{N}(v)} 3f(d_i) - \sum_{i \in \mathcal{N}(u)} f(d_i) - \sum_{i \in \mathcal{N}(v)} f(d_i)$$

$$= 2f(d_u) + 2f(d_v) + \left( \frac{3}{\mathsf{Jaccard}_f^{-1}(\mathcal{N}(u), \mathcal{N}(v)) + 1} - 1 \right) \left( \sum_{i \in \mathcal{N}(u)} f(d_i) + \sum_{i \in \mathcal{N}(v)} f(d_i) \right)$$

$$= 2f(d_u) + 2f(d_v) + \left( \frac{3\,\mathsf{Jaccard}_f(\mathcal{N}(u), \mathcal{N}(v))}{1 + \mathsf{Jaccard}_f(\mathcal{N}(u), \mathcal{N}(v))} - 1 \right) \left( \sum_{i \in \mathcal{N}(u)} f(d_i) + \sum_{i \in \mathcal{N}(v)} f(d_i) \right)$$

$$= 2f(d_u) + 2f(d_v) + \left( 2 - \frac{3}{1 + \mathsf{Jaccard}_f(\mathcal{N}(u), \mathcal{N}(v))} \right) \left( \sum_{i \in \mathcal{N}(u)} f(d_i) + \sum_{i \in \mathcal{N}(v)} f(d_i) \right)$$

## B Related work

Alon & Yahav (2020) first proposed that information flowing through a graph can be over-squashed due to bottleneck structures. This idea was quickly recognized by the graph learning community, and over-squashing has become one of the core challenges faced by subsequent researchers designing new deep graph models. The next notable breakthrough in over-squashing research came from Topping et al. (2021), who demonstrated the use of the discrete decrement as an upper bound on the Jacobian matrix norm, thus transforming the ambiguous problem of determining over-squashed edges into a precisely defined curvature calculation.

Since then, discrete curvature has garnered significant interest within the graph learning community. One of the most typical approaches is curvature-based graph rewiring, whose core idea involves performing curvature-based preprocessing on graph data before executing graph deep learning to eliminate edges with extremely high or low curvature. Representative methods include those by SDRF (Topping et al., 2021), BORF (Nguyen et al., 2023), SJLR (Giraldo et al., 2023), AFR-3 (Fesser & Weber, 2024), among others. Another approach involves directly integrating curvature into end-to-end GNN models. For instance, Li et al. (2022) utilized discrete curvature to improve aggregation weights in message passing; Sun et al. (2022) employed a hierarchical attention mechanism based on mixed-curvature spaces to capture complex graph structures and enhance performance; Fu et al. (2025) et al. proposed a curvature-optimized variational information bottleneck principle to optimize information transmission on graphs; while Chen et al. (2025) designed a continuous-depth GNN with effects similar to rewiring methods via curvature Ricci flow. In addition, there is another type of method that proposes the asynchronous message passing mechanism (Gutteridge et al., 2023; Chen et al., 2024; Bose & Das, 2025). Their core innovation lies in combining the above two categories: graph reconnection and end-to-end graph network.

In terms of theoretical progress, Di Giovanni et al. (2023a) discussed how factors such as network width, depth, and graph topology affect over-squeezing with the help of arrival time; Di Giovanni et al. (2023b) studied the relationship between over-squeezing and expressiveness; Nguyen et al. (2023) et al. established a unified understanding of over-smoothing and over-squeezing through ORC; Chen et al. (2024) gave the conditions for GNN to exhibit local priority through curvature. Tori et al. (2024b) proposed that the effectiveness of curvature-based graph rewiring methods may be due to the correction of outliers.

## C    COMPUTE MOSR

In this paper, we introduce the metric $MOSR_q$, which measures the proportion of over-squashed edges that are ignored by the discrete curvature. MOSR is calculated using the Jacobian matrix norm, making it a tightly coupled metric with the model. This section details the calculation of MOSR.

To ensure fairness, we want the network model to be optimal when calculating the Jacobian matrix norm. To achieve this, we used Optuna Akiba et al. (2019) to perform 200 times hyperparameter search for each model on each dataset, with the hyperparameter search range fixed uniformly (as shown in Table 5. The hyperparameter search results are shown in Table 6, Table 7 and Table 8.

We then randomly initialize the GNN with the optimal hyperparameter combination and calculate the Jacobian matrix norm $\left\| \frac{\partial \mathbf{h}_j^{(L)}}{\partial \mathbf{h}_i^{(0)}} \right\|$ for each edge $(i, j)$ in the dataset via the autograd module in pytorch. We directly use randomly initialized GNNs without any training (See Appendix D.1 for more discussion on this).

Following the definition of discrete curvature, we compute a curvature value for each edge and then compute MOSR as defined in Section 4. We repeat each experiment 10 times with different initialization and report the average value. Our experimental platform is Intel(R) Xeon(R) Gold 6240C CPU @ 2.60GHz and NVIDIA GeForce RTX 4090 × 4. We set the resource consumption limit for a single experiment to 12 hours of runtime and 24GB of GPU memory. Experiments exceeding this limit will be marked as Out of Resources (OOR).

Table 5: Scope of hyperparameter search

| Hyperparameter | Range |
|---|---|
| n_layer | [2,3,4,5] |
| n_hidden | [64, 128, 256] |
| dropout | [0, 0.1, 0.2, 0.3, 0.4, 0.5, 0.6, 0.7] |
| lr | [0.01, 0.005, 0.002, 0.001, 0.0005] |
| weight_decay | [0.005, 0.001, 0.0005, 0.0001, 0] |
| norm | ["batch_nrom", "layer_norm", none] |

Table 6: The optimal hyperparameter combination of GCN on different datasets

|  | n_layer | n_hidden | dropout | lr | weight_decay | norm |
|---|---|---|---|---|---|---|
| **Cora** | 2 | 64 | 0.7 | 0.001 | 0.005 | none |
| **Citeseer** | 2 | 256 | 0.5 | 0.001 | 0.005 | none |
| **Pubmed** | 2 | 64 | 0.6 | 0.005 | 0.0001 | none |
| **Computers** | 2 | 256 | 0.3 | 0.002 | 0.0001 | none |
| **Photo** | 2 | 256 | 0.5 | 0.0005 | 0 | ln |
| **CS** | 2 | 128 | 0.7 | 0.0005 | 0.001 | none |
| **Physics** | 2 | 64 | 0.1 | 0.005 | 0.001 | none |
| **WikiCS** | 2 | 256 | 0.5 | 0.0005 | 0 | ln |
| **Cora_ML** | 5 | 64 | 0.6 | 0.001 | 0.0001 | none |
| **Cora_Full** | 2 | 128 | 0.5 | 0.01 | 0.001 | none |
| **DBLP** | 3 | 256 | 0.3 | 0.005 | 0.005 | none |
| **Cornell** | 2 | 64 | 0.4 | 0.0005 | 0.005 | none |
| **Texas** | 2 | 256 | 0.2 | 0.002 | 0.0005 | none |
| **Wisconsin** | 3 | 128 | 0.2 | 0.01 | 0.001 | bn |
| **Chameleon** | 2 | 64 | 0.6 | 0.001 | 0.0005 | none |
| **Squirrel** | 4 | 128 | 0.2 | 0.01 | 0.0001 | bn |
| **Roman-empire** | 2 | 256 | 0.1 | 0.002 | 0.0001 | none |
| **Tolokers** | 2 | 256 | 0.7 | 0.01 | 0 | ln |
| **Questions** | 2 | 256 | 0.6 | 0.01 | 0.001 | ln |
| **Amazon-ratings** | 2 | 256 | 0.2 | 0.01 | 0 | ln |
| **Minesweeper** | 3 | 256 | 0.5 | 0.005 | 0 | ln |

Table 7: The optimal hyperparameter combination of GAT on different datasets

|  | n_layer | n_hidden | dropout | lr | weight_decay | norm |
|---|---|---|---|---|---|---|
| **Cora** | 3 | 128 | 0.6 | 0.0005 | 0.005 | none |
| **Citeseer** | 2 | 64 | 0.4 | 0.005 | 0.005 | none |
| **Pubmed** | 2 | 64 | 0.5 | 0.0005 | 0 | bn |
| **Computers** | 4 | 128 | 0.5 | 0.001 | 0.001 | ln |
| **Photo** | 3 | 128 | 0.5 | 0.005 | 0.0001 | none |
| **CS** | 2 | 256 | 0.6 | 0.01 | 0.0001 | none |
| **Physics** | 2 | 256 | 0.7 | 0.002 | 0.001 | ln |
| **WikiCS** | 2 | 256 | 0.6 | 0.0005 | 0.0001 | bn |
| **Cora_ML** | 5 | 64 | 0.7 | 0.0005 | 0 | ln |
| **Cora_Full** | 2 | 256 | 0.7 | 0.01 | 0.005 | bn |
| **DBLP** | 3 | 64 | 0.5 | 0.01 | 0.005 | none |
| **Cornell** | 4 | 64 | 0.4 | 0.005 | 0.005 | bn |
| **Texas** | 5 | 256 | 0.7 | 0.002 | 0 | ln |
| **Wisconsin** | 4 | 128 | 0.7 | 0.01 | 0.005 | none |
| **Chameleon** | 5 | 128 | 0.3 | 0.0005 | 0.005 | ln |
| **Squirrel** | 2 | 256 | 0.6 | 0.002 | 0.001 | bn |
| **Roman-empire** | 2 | 128 | 0.3 | 0.005 | 0 | none |
| **Tolokers** | 3 | 256 | 0.6 | 0.005 | 0.0001 | none |
| **Questions** | 4 | 256 | 0.3 | 0.001 | 0.0001 | none |
| **Amazon-ratings** | 2 | 128 | 0.1 | 0.0005 | 0.0001 | bn |
| **Minesweeper** | 3 | 128 | 0.4 | 0.01 | 0.0001 | none |

Table 8: The optimal hyperparameter combination of GraphSAGE on different datasets

|  | n_layer | n_hidden | dropout | lr | weight_decay | norm |
|---|---|---|---|---|---|---|
| **Cora** | 2 | 64 | 0.5 | 0.0005 | 0.005 | none |
| **Citeseer** | 2 | 128 | 0.2 | 0.005 | 0.005 | none |
| **Pubmed** | 2 | 256 | 0.5 | 0.005 | 0.001 | none |
| **Computers** | 4 | 256 | 0.7 | 0.0005 | 0.0001 | none |
| **Photo** | 2 | 64 | 0.6 | 0.002 | 0 | none |
| **CS** | 2 | 256 | 0.6 | 0.001 | 0.0001 | none |
| **Physics** | 3 | 256 | 0.7 | 0.005 | 0.0001 | ln |
| **WikiCS** | 3 | 256 | 0.7 | 0.005 | 0 | ln |
| **Cora_ML** | 3 | 256 | 0.1 | 0.001 | 0.0005 | none |
| **Cora_Full** | 2 | 256 | 0.2 | 0.0005 | 0.005 | ln |
| **DBLP** | 5 | 64 | 0.7 | 0.01 | 0.0005 | bn |
| **Cornell** | 3 | 64 | 0.1 | 0.0005 | 0.001 | none |
| **Texas** | 2 | 64 | 0.2 | 0.001 | 0.005 | none |
| **Wisconsin** | 3 | 64 | 0.3 | 0.01 | 0.001 | none |
| **Chameleon** | 2 | 128 | 0.7 | 0.01 | 0 | none |
| **Squirrel** | 3 | 256 | 0.7 | 0.005 | 0 | ln |
| **Roman-empire** | 4 | 256 | 0.5 | 0.01 | 0 | bn |
| **Tolokers** | 4 | 64 | 0.5 | 0.005 | 0 | bn |
| **Questions** | 5 | 128 | 0.7 | 0.0005 | 0.0005 | none |
| **Amazon-ratings** | 3 | 256 | 0.5 | 0.002 | 0 | bn |
| **Minesweeper** | 5 | 256 | 0.4 | 0.001 | 0.0005 | none |

# D    SUPPLEMENTARY EXPERIMENTS

## D.1    EFFECTS OF TRAINING AND $q$ ON MOSR.

We explore the impact of the number of training rounds of the GNN model and the choice of $p$ value on the value of MOSR. Our experimental settings cover different models, datasets, and curvature definitions.

> • **Observation 9**: As the number of training epochs increases, the value of MOSR will slowly increase and gradually stabilize as the training converges (because the difference in MOSR between epoch=200 and epoch=500 is significantly smaller than the difference in MOSR between epoch=0 and epoch=50).
>
> • **Observation 10**: When $p$ is smaller, the criteria for being considered an over-squashing edge will be stricter, and the value of $\text{MOSR}_p$ will also become smaller.
>
> • **Observation 11**: Observation 9 and observation 10 hold for all discrete curvatures (compare Figure 5 with Figures 6 and 7), all models (compare Figure 5 with Figures 8 and 9), and all dataset (compare Figure 5 with Figures 10 and 11).

The MOSR values reported in the main paper are all based on an untrained model (epoch=0) and small $p$-values (10 and 25). Based on the above observations, this means that we are reporting a **lower bound** on the probability that the discrete curvature ignores over-squashing edges. Under any other settings (e.g., more epochs and higher $p$), the discrete curvature will ignore even more over-squashing edges than we report.

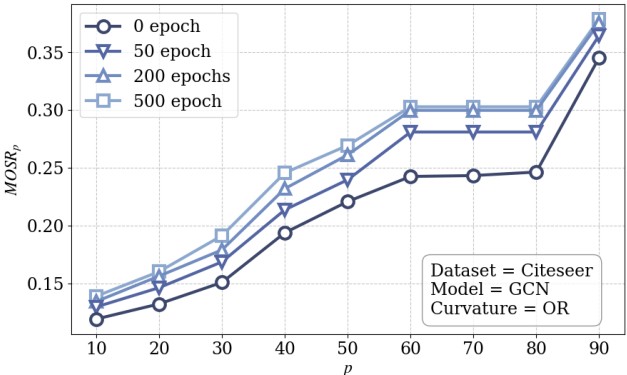

Figure 5: When the dataset is Citeseer, the model is GCN, and the discrete curvature is Ollivier Ricci Curvature, the impact of different training epochs and $p$-values on MOSR.

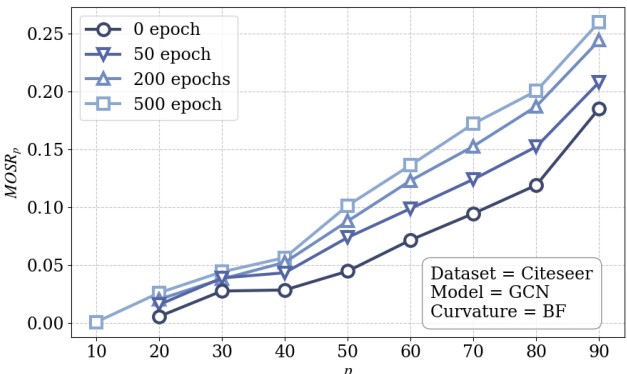

Figure 6: When the dataset is Citeseer, the model is GCN, and the discrete curvature is balanced Forman Curvature, the impact of different training epochs and $p$-values on MOSR.

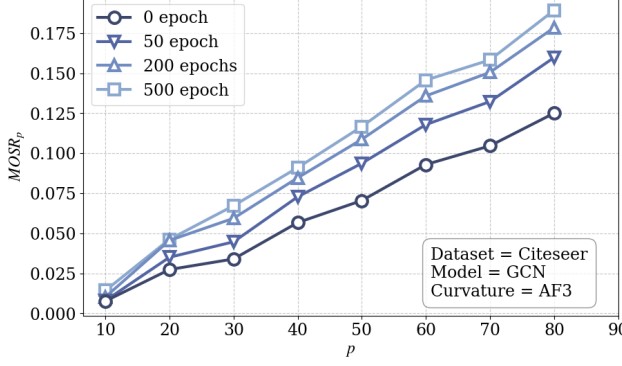

Figure 7: When the dataset is Citeseer, the model is GCN, and the discrete curvature is augmented Forman-3 Curvature, the impact of different training epochs and $p$-values on MOSR.

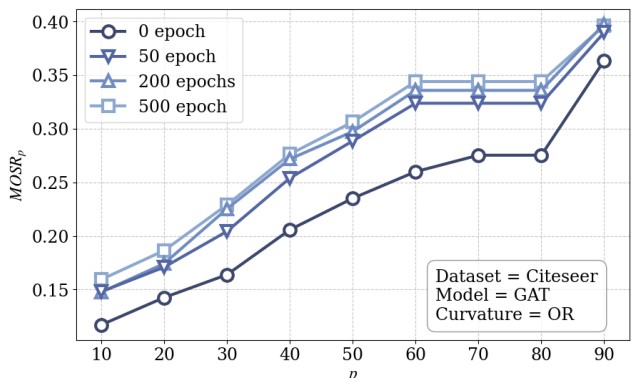

Figure 8: When the dataset is Citeseer, the model is GAT, and the discrete curvature is Ollivier Ricci Curvature, the impact of different training epochs and $p$-values on MOSR.

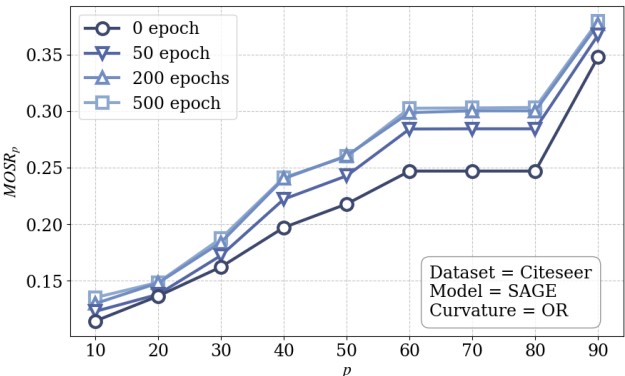

Figure 9: When the dataset is Citeseer, the model is GraphSAGE, and the discrete curvature is Ollivier Ricci Curvature, the impact of different training epochs and $p$-values on MOSR.

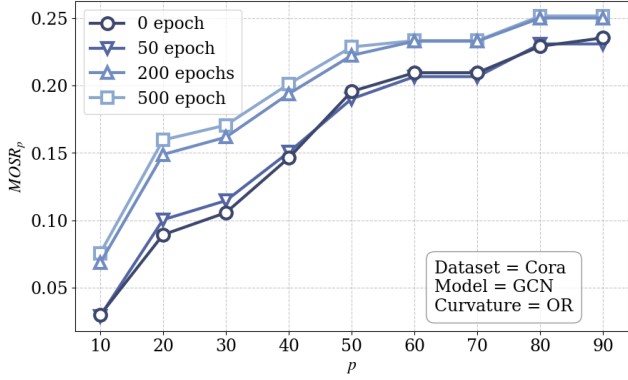

Figure 10: When the dataset is Cora, the model is GCN, and the discrete curvature is Ollivier Ricci Curvature, the impact of different training epochs and $p$-values on MOSR.

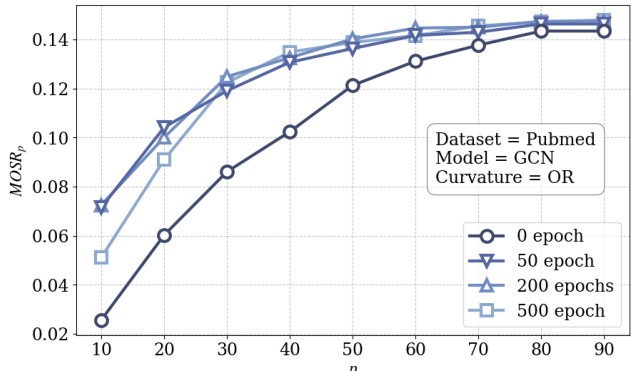

Figure 11: When the dataset is Pubmed, the model is GCN, and the discrete curvature is Ollivier Ricci Curvature, the impact of different training epochs and $p$-values on MOSR..

## D.2 WEIGHTING FUNCTION

Assuming that node $u$ and node $v$ have a common node $i$, for an MPNN as shown in Equation (1), the norm of the Jacobian matrix of $u$'s information flowing through $i$ to $v$ is:

$$
\left\| \frac{\partial \mathbf{h}_v^{(l+2)}}{\partial \mathbf{h}_i^{(l+1)}} \cdot \frac{\partial \mathbf{h}_i^{(l+1)}}{\partial \mathbf{h}_u^{(l)}} \right\|_F
$$

$$
= \rho \left\| \mathbf{W}^{(l+1)} \mathbf{W}^{(l)} \left( (\mathbf{D}+\mathbf{I})^{-1/2}(\mathbf{A}+\mathbf{I})(\mathbf{D}+\mathbf{I})^{-1/2} \right)_{i,v} \left( (\mathbf{D}+\mathbf{I})^{-1/2}(\mathbf{A}+\mathbf{I})(\mathbf{D}+\mathbf{I})^{-1/2} \right)_{u,i} \right\|_F
$$

$$
= \rho \left\| \mathbf{W}^{(l+1)} \mathbf{W}^{(l)} \right\|_F \cdot \underbrace{\frac{1}{1+d_i}}_{\text{The influence of intermediate node i}} \cdot \frac{1}{\sqrt{(1+d_u)(1+d_v)}}.
$$

Therefore, we set $f(d_i)$ to $1/(1+d_i)$ to balance the contribution of any first-order neighbor node in computing the discrete curvature. We also explore the effects of weight functions that decay faster or slower than $1/(1+d_i)$ (i.e., $(1+d_i)^{-2}$ and $(1+d_i)^{-1/2}$) in Table 9.

> • **Observation 12**: On average, when $f(d) = (1+d)^{-1}$, the value of MOSR is generally slightly smaller (better) than when $f(d) = (1+d)^{-2}$ or $f(d) = (1+d)^{-1/2}$.

## D.3 CURVATURE-BASED GRAPH LEARNING

We explore the impact of different discrete curvature definitions on graph learning. We select Stochastic Discrete Ricci Flow (SDRF, (Topping et al., 2021)) as a representative graph rewiring method and Graph Neural Ricci Flow (GNRF, (Chen et al., 2025)) as a representative end-to-end method. The results are shown in Tables 10 and 11, respectively.

> • **Observation 13**: In the graph rewiring method SDRF, WAF3 (and its approximations) outperform other curvatures on two datasets; and in the end-to-end GNRF, they outperform other curvatures on all three datasets.
>
> • **Observation 14**: When hash=100, approximately WAF3 performs as well as or better than balanced Forman curvature and Ollivier Ricci curvature on both SDRF and GNRF.

The above observations show that WAF3 can achieve better performance in curvature-based graph learning than previous curvature methods, and can do well enough even with rough approximations.

Table 9: The impact of different attenuation functions on the calculation of WAF3. The values in the table represent $MOSR_{10}$ and $MOSR_{25}$.

| | $f(d) = (1+d)^{-2}$ | | | $f(d) = (1+d)^{-1/2}$ | | | $f(d) = (1+d)^{-1}$ | | |
|---|---|---|---|---|---|---|---|---|---|
| | GCN | GAT | SAGE | GCN | GAT | SAGE | GCN | GAT | SAGE |
| **Cora** | .009/.056 | .169/.178 | .196/.196 | .000/.016 | .147/.154 | .172/.172 | .000/.014 | .157/.166 | .183/.183 |
| **Citeseer** | .044/.092 | .223/.235 | .286/.286 | .007/.026 | .205/.205 | .255/.273 | .020/.040 | .210/.216 | .279/.280 |
| **Pubmed** | .008/.013 | .023/.023 | .025/.025 | .000/.001 | .009/.010 | .011/.011 | .001/.002 | .013/.014 | .015/.015 |
| **Computers** | .014/.017 | .018/.018 | .018/.018 | .001/.004 | .009/.009 | .010/.010 | .002/.005 | .011/.011 | .011/.011 |
| **Photo** | .020/.025 | .026/.026 | .026/.026 | .029/.030 | .033/.033 | .033/.033 | .021/.024 | .027/.027 | .027/.027 |
| **CS** | .070/.079 | .092/.093 | .094/.094 | .011/.021 | .068/.068 | .069/.069 | .009/.034 | .075/.075 | .076/.076 |
| **Physics** | .060/.069 | .071/.072 | OOR | .019/.023 | .049/.053 | OOR | .020/.040 | .055/.058 | OOR |
| **WikiCS** | .065/.069 | .070/.071 | .071/.071 | .203/.203 | .205/.205 | .205/.205 | .140/.141 | .144/.144 | .144/.144 |
| **Cora_ML** | .074/.085 | .115/.119 | .119/.119 | .020/.048 | .088/.090 | .091/.093 | .025/.056 | .098/.103 | .103/.103 |
| **Cora_Full** | OOR | OOR | OOR | OOR | OOR | OOR | OOR | OOR | OOR |
| **DBLP** | .056/.069 | .084/.086 | .088/.092 | .002/.010 | .034/.037 | .039/.043 | .020/.022 | .052/.055 | .057/.059 |
| **Cornell** | .000/.000 | .127/.138 | .141/.156 | .000/.000 | .120/.124 | .143/.143 | .000/.000 | .116/.122 | .138/.143 |
| **Texas** | .000/.000 | .132/.132 | .140/.143 | .000/.000 | .118/.123 | .128/.137 | .000/.000 | .134/.139 | .143/.149 |
| **Wisconsin** | .000/.000 | .122/.132 | .130/.130 | .000/.000 | .102/.109 | .116/.126 | .000/.000 | .108/.117 | .122/.134 |
| **Chameleon** | .037/.043 | .040/.044 | .044/.044 | .090/.091 | .100/.101 | .104/.104 | .065/.066 | .075/.076 | .079/.079 |
| **Squirrel** | .015/.015 | .015/.015 | .015/.015 | .103/.103 | .100/.100 | .100/.101 | .039/.039 | .041/.041 | .041/.041 |
| **Roman-empire** | .000/.002 | .293/.432 | .452/.454 | .000/.004 | .352/.431 | .452/.453 | .000/.001 | .351/.431 | .453/.453 |
| **Tolokers** | .007/.007 | OOR | .007/.007 | .001/.001 | OOR | .002/.002 | .002/.002 | OOR | .003/.003 |
| **Questions** | .011/.018 | .031/.031 | .032/.032 | .001/.006 | .016/.016 | .018/.018 | .003/.011 | .023/.024 | .025/.025 |
| **Amazon-ratings** | .194/.219 | .201/.209 | .223/.224 | .178/.235 | .240/.248 | .261/.263 | .159/.218 | .223/.231 | .245/.246 |
| **Minesweeper** | .270/.270 | .271/.271 | .271/.271 | .233/.233 | .234/.234 | .233/.233 | .191/.192 | .192/.192 | .192/.192 |
| **Average** | **.048/.057** | **.118/.122** | **.125/.126** | **.045/.053** | **.117/.124** | **.128/.129** | **.036/.045** | **.111/.118** | **.123/.124** |

Table 10: Accuracy on downstream classification tasks after graph rewiring using SDRF (Topping et al., 2021) with different curvature definitions. The experimental setup for this experiment remains identical to the original paper. *Indicates data referenced from Topping et al. (2021).

| | Cornell | Texas | Wisconsin |
|---|---|---|---|
| Balanced Forman | $57.54 \pm 0.34^*$ | $70.35 \pm 0.60^*$ | $61.55 \pm 0.84^*$ |
| Ollivier Ricci | $55.56 \pm 1.05$ | $64.51 \pm 0.26$ | $58.51 \pm 0.60$ |
| Augmented Forman 3 | $\mathbf{58.29 \pm 0.92}$ | $73.29 \pm 0.64$ | $63.19 \pm 0.96$ |
| Weighted AF3 | $57.91 \pm 0.54$ | $\mathbf{73.62 \pm 0.62}$ | $65.64 \pm 0.24$ |
| Approximately Weighted AF3 (100 hash) | $58.21 \pm 0.64$ | $71.96 \pm 0.68$ | $63.74 \pm 0.34$ |
| Approximately Weighted AF3 (1000 hash) | $57.88 \pm 0.78$ | $70.55 \pm 0.61$ | $65.10 \pm 0.63$ |
| Approximately Weighted AF3 (10000 hash) | $57.67 \pm 0.62$ | $72.11 \pm 0.51$ | $\mathbf{65.99 \pm 0.67}$ |

Table 11: The accuracy of end-to-end model GNRF (Chen et al., 2025) on downstream classification tasks using different curvature definitions. The experimental setup for this experiment remains exactly the same as the original GNRF paper.

| | Cornell | Texas | Wisconsin |
|---|---|---|---|
| Balanced Forman | $84.37 \pm 3.11$ | $83.15 \pm 6.25$ | $83.15 \pm 3.25$ |
| Ollivier Ricci | $81.26 \pm 5.45$ | $79.95 \pm 5.14$ | $81.26 \pm 2.36$ |
| Augmented Forman 3 | $84.21 \pm 5.26$ | $85.66 \pm 3.25$ | $79.26 \pm 1.59$ |
| Weighted AF3 | $84.62 \pm 3.26$ | $\mathbf{87.11 \pm 1.03}$ | $84.66 \pm 0.34$ |
| Approximately Weighted AF3 (100 hash) | $84.16 \pm 4.10$ | $84.26 \pm 2.34$ | $82.56 \pm 1.26$ |
| Approximately Weighted AF3 (1000 hash) | $83.12 \pm 2.16$ | $85.79 \pm 1.26$ | $\mathbf{85.11 \pm 1.26}$ |
| Approximately Weighted AF3 (10000 hash) | $\mathbf{84.99 \pm 2.13}$ | $86.35 \pm 2.11$ | $84.25 \pm 0.67$ |

# E    USE OF LLMS

In this project, we only used LLMs to find and correct grammatical errors and polish the text.

