# OpenReview forum: "Rethinking the Gold Standard: Why Discrete Curvature Fails to Fully Capture Over-squashing in GNNs?"
_ICLR.cc/2026/Conference — ICLR 2026 Poster_

### Official Review · Reviewer_QN1W · 2025-10-23

**Soundness:** 3
**Presentation:** 3
**Contribution:** 3
**Rating:** 6
**Confidence:** 4

**Summary:**

The paper challenges the common assumption that discrete curvature is a reliable proxy for detecting over-squashing in GNNs, showing via a family of counterexample graphs and formal results that many severely squashed edges can still have positive curvature—so high negative curvature is sufficient but not necessary for over-squashing. To quantify this gap, the authors introduce MOSR, a metric that counts the fraction of truly over-squashed edges missed by a curvature rule; across models and 21 datasets, Ollivier–Ricci misses roughly 30–40% of such edges, and a betweenness analysis indicates that curvature mostly finds “bridge” edges while often ignoring in-cluster bottlenecks. They then propose Weighted Augmented Forman-3 (WAF3), a degree-weighted refinement that theoretically avoids the counterexamples and empirically reduces MOSR relative to prior curvatures while retaining low complexity. Finally, they derive an equivalent weighted-Jaccard form and a MinHash-based approximation that lowers runtime to linear in the number of edges, enabling curvature computation on graphs with about 5 million edges

**Strengths:**

The paper is original in challenging a widely assumed link between discrete curvature and over-squashing and in formalizing how and why common curvatures can fail, complemented by a clear diagnostic (MOSR) that quantifies misses rather than relying on anecdotal examples. It shows solid technical quality through explicit counterexamples, theoretical guarantees for the proposed WAF3 variant, and careful complexity considerations, including an equivalent weighted-Jaccard view and a MinHash-based approximation that makes large-scale computation practical. The presentation is clear: definitions, constructions, and algorithms are spelled out with enough detail to reproduce both the counterexamples and the fast estimator. Significance is high for both researchers and practitioners who use curvature to guide model design or interventions for over-squashing, since the work both tempers over-reliance on existing curvatures and supplies a drop-in alternative that is faster and empirically more aligned with actual over-squashing.

**Weaknesses:**

Despite strong theoretical framing, the empirical story rests heavily on MOSR—a model- and label-dependent proxy for “true” over-squashing—so it remains unclear how much the proposed curvature actually improves downstream GNN behavior (e.g., training stability, accuracy, or robustness) beyond ranking edges; adding intervention studies (rewiring, weighting, or attention bias guided by WAF3 vs prior curvatures) and showing consistent end-task gains would strengthen the claim. The novelty of WAF3 also needs sharper positioning: prior work on curvature-based structural encodings (e.g., “Effective structural encodings via local curvature profiles”) should be cited in the introduction and connections to other works that use (weighted) augmented Forman–Ricci variants (e.g., “Augmentations of Forman’s Ricci curvature and their applications in community detection”) should be cited and discussed.

**Questions:**

Could you provide intervention-based validation showing that using WAF3 to guide rewiring, edge weighting, or attention bias leads to consistent gains in downstream GNN performance versus Ollivier–Ricci and prior Forman variants, with compute-matched baselines, multiple seeds, and analyses of where improvements concentrate (e.g., non-bridge bottlenecks)?

---

> ### Author Response · Authors · 2025-11-21
>
> We acknowledge the hard work the reviewers put into providing professional and detailed feedback, and below is our detailed response.
>
>
> > it remains unclear how much the proposed curvature actually improves downstream GNN behavior.
>
> In Appendix D.3, we analyzed the performance of WAF3 and its approximations on downstream tasks, including graph rewiring methods and end-to-end depth graph models. The results show that our proposed curvature outperforms existing algorithms in the vast majority of cases. But besides that, we want to emphasize two points.
>
> 1. Our paper focuses on improving the definition of curvature rather than enhancing the performance of downstream tasks. And **the progress of a metric is not necessarily related to the progress of methods based on that metric**. Graph rewiring algorithms are a good example of this. The first graph rewiring algorithm was proposed by Topping et al [1]. In their paper, they envisioned that graph rewiring algorithms could improve the performance of downstream tasks by alleviating over-squashing. However, a recent study [2] pointed out a surprising point: the effectiveness of graph rewiring does not actually stem from alleviating over-squashing, but from reducing outliers. This fundamentally overturns researchers' understanding of graph rewiring. Our paper proposes a definition of discrete edge curvature that is more effective and more efficient for detecting over-quashing edges, but this does not mean that it is more suitable for anomaly detection. This is just one example of graph rewiring, and we do not yet know whether other types of algorithms/methods also have similar mechanisms that have not yet been understood by researchers.
>
> 2. Many works focus on improving the performance of downstream graph learning tasks based on curvature, but few discuss how to accelerate the computation of graph curvature. Our work makes significant progress in this regard. Existing graph curvature-based work can leverage our proposed WAF3 based on the MinHash approximation to significantly improve computational speed without weakening (or even enhancing) the performance of downstream tasks. Therefore, we believe that our work makes a unique contribution to this field.
>
>
> > prior work on curvature-based structural encodings (e.g., “Effective structural encodings via local curvature profiles”) should be cited in the introduction and connections to other works that use (weighted) augmented Forman–Ricci variants (e.g., “Augmentations of Forman’s Ricci curvature and their applications in community detection”) should be cited and discussed.
>
>
> Thank you for your reminder. We have appropriately cited these relevant and classic works in the introduction and discussed in the related works sections.
>
>
> > Could you provide intervention-based validation showing that using WAF3 to guide rewiring, edge weighting, or attention bias leads to consistent gains in downstream GNN performance versus Ollivier–Ricci and prior Forman variants, with compute-matched baselines, multiple seeds, and analyses of where improvements concentrate (e.g., non-bridge bottlenecks)?
>
>
> As mentioned above, we conducted similar experiments in Appendix D.3. While the experimental results favored WAF3, we believe this is not a logically sound way to demonstrate WAF3's superiority. Similarly, while WAF3 effectively identifies bottlenecks within clusters, this does not necessarily mean that the performance of nodes within clusters will be significantly improved in downstream tasks. For example, in graph rewiring, if the real reason the algorithm works is to eliminate outliers, then identifying bottlenecks within a cluster may not provide much benefit, since outliers are outside the cluster rather than inside it. Furthermore, in node classification tasks, the accuracy of nodes within clusters is often higher than that of nodes located at the cluster edges (or on bridges). Even if a better definition of curvature improves the performance of downstream tasks, due to marginal effects, we cannot infer that this is primarily due to the increased accuracy of nodes within clusters.
>
>
> [1] Understanding over-squashing and bottlenecks on graphs via curvature
>
> [2] The Effectiveness of Curvature-Based Rewiring and the Role of Hyperparameters in GNNs Revisited

---

### Official Review · Reviewer_jUf2 · 2025-10-27

**Soundness:** 2
**Presentation:** 3
**Contribution:** 2
**Rating:** 4
**Confidence:** 4

**Summary:**

The paper argues that having a negative discrete curvature (e.g., Ollivier–Ricci, Forman variants) is not a necessary condition for over-squashing in GNNs. It provides a counterexample family Gnm, showing edges can be heavily over-squashed while having positive curvature; introduces MOSR (Missed Over-Squashing Ratio) to quantify how many over-squashed edges are missed by curvature; proposes a new curvature WAF3 (Weighted Augmented Forman-3) with a degree-weighting; and develops a MinHash-based approximation to scale WAF3. Experiments report MOSR across 21 datasets and three architectures (GCN/GAT/GraphSAGE), plus efficiency and rank-correlation results for the approximation.

**Strengths:**

- This paper shows that negative curvature does not imply necessity, with a concrete counterexample and theorem.
- MOSR is a reasonable way to quantify “missed” oversquashed edges.
- WAF3 is intuitive, keeps low complexity, and is shown to remove the counterexample under mild conditions.
- Equivalent form via weighted Jaccard + MinHash is clever and gives large speedups with good rank fidelity.

**Weaknesses:**

- Eqn (1) is not the MPNN model; that’s the GCN model. MPNN is a generalization of GCN. This is problematic since most of the results in over-squashing in the previous literature are for MPNNs, while in this paper, they’re only true for GCNs. This reduces the scope and impact of the proposed work and diminishes the results in Table 4, since the theoretical understanding is only available for GCN.
- Similarly, multiple formal statements (Lemma 2, Theorem 5) say “Assume an L-layer MPNN as in (1).” But Eq. (1) is the symmetric-normalized GCN layer with ReLU; it is not a general MPNN. The proofs that rely on the exact matrix M therefore apply to GCN-style propagation; generalizations would require additional conditions on the aggregation kernels.
- The last part of Section 4 lacks clear intuition on how over-squashing relates to bottlenecks and how this differs from “bridges”. Prior theory (e.g., Topping et al.) does not imply that removing bottlenecks eliminates over-squashing, yet the paper does not provide a concrete narrative or example to reconcile this. In particular, the manuscript does not illustrate what an intra-cluster over-squashed edge looks like (low betweenness, still over-squashed) or explain the mechanisms that make it over-squashed despite no obvious bottleneck. This gap makes the conceptual contribution harder to follow and weakens the takeaways of Section 4.
- The choice of the Forman-3 curvature for the extension seems arbitrary. Indeed, the balance Forman w/o 4-cycle and the Jost-Liu Forman have the same complexity as the Augmented Forman-3 according to Table 1. However, those are not compared in Table 2, and we don’t know if these other two computationally efficient metrics are better or worse than the Forman-3 curvature.
- I understood the reasoning behind Figure 4, but I feel it’s also important to compare the efficient version of the proposed curvature in terms of MOSR, as in Table 4. This new version might be more efficient, but the main point of this paper is to avoid missing the over-squashed edges where the curvature is positive, and if this is not the case for the efficient formulation, why should one go with this alternative? If the important point is the ordering (I agree with the authors on this), why should one choose this approach over something that achieves the correct rank, while being perhaps more efficient?


**General comment**: This is a timely paper, but two pieces are missing to make the contribution fully convincing. (1) The manuscript does not clearly connect the proposed most efficient curvature variant to the central claim of the paper (reducing missed over-squashed edges). Please make the causal chain explicit: why this efficient metric preserves the ordering or detection properties that matter for oversquashing, and how that supports the paper’s main thesis. (2) The paper lacks a large-scale, real-world evaluation demonstrating impact. A straightforward way to address this is to plug the efficient curvature into an existing rewiring pipeline and report results on a sparse OGB node property benchmark—e.g., ogbn-products or ogbn-papers100M—including both accuracy and compute (runtime/memory). This would substantiate scalability claims and show that the efficiency gains translate into practical benefits without undermining effectiveness.


**Minor comments:**

- Line 58: “This theorem…”, which theorem? At this point in the paper, it’s not clear this is supported by a theorem.
- Please define acronyms the first time they’re used, e.g., MOSR.
- It seems like Table 3 has not been referenced in the paper.
- Figure 2 is violating the margins.
- Typo in Theorem 5? There dose not.
- Could you include the statistics of the datasets? I think this is important to put things in context.

**Questions:**

- What is \rho in Lemma 2? This also appears in Appendix A.1, but I’m not sure what this is.
- Assumption 1 is not clear. What does the paper mean by all paths in the computation graph? What’s the computation graph in this case?
- The notation in Definition 3 is not clear. N_1 and N_2 are the 1-hop and 2-hop neighborhoods of whom?
- s_q is the proportion of over-squashing edges that are not identified by curvature, but I don’t see this in the equation. Do the authors mean MOSR_q?
- What’s the number of layers L for the experiments in Table 2?
- Could the authors provide the same results as in Table 2 for the balance Forman w/o 4-cycle and the Jost-Liu Forman? These two curvature metrics are also computationally efficient.
- In Appendix A.1, Lemma 5 is used to get the bound, but I didn’t find this Lemma. Is it Lemma 7? Even if this is the case, I’m not quite sure how to get the bound in the first Equation on page 14 (please label the equations). Why is greater than or equal to? I’m confused at that point.

---

> ### Author Response · Authors · 2025-11-15
> **(1/N) About General Comment (1)**
>
> We are very grateful to the reviewers for providing such detailed comments, and we believe they spent a great deal of time on this. We will provide detailed rebuttals point by point below, ordered by importance.
>
> **General Comment (1)**
>
> > The manuscript does not clearly connect the proposed most efficient curvature variant to the central claim of the paper (reducing missed over-squashed edges). Please make the causal chain explicit: why this efficient metric preserves the ordering or detection properties that matter for oversquashing, and how that supports the paper’s main thesis.
>
> In Section 5, we demonstrate in detail that $\mathsf{WAF3}$ does indeed identify over-squashing edges more effectively than existing discrete curvature, both theoretically (Theorem 5) and experimentally (Table 4). The construction of $\mathsf{WAF3}$ stems from our analysis of the causes of over-squashing edge identification failures in counterexamples: For an edge $u\sim v$, the unrelated nodes (i.e., ${\mathcal N}_2$ in Definition 3 and Figure 1) connected to the shared first-order nodes of $u$ and $v$ (i.e., ${\mathcal N}_1$ in Definition 3 and Figure 1) will "dilute" the propagation of information between $u$ and $v$. In $\mathsf{WAF3}$, we appropriately adjust the weights of these nodes in ${\mathcal N}_1$ according to their degree, thereby eliminating the influence of nodes in ${\mathcal N}_2$, which leads to a better definition of discrete curvature.
>
> $\widehat{\mathsf{WAF3}}$ is an approximate version of $\mathsf{WAF3}$ built based on the MinHash algorithm. Note that the MinHash algorithm theoretically guarantees that as long as the sampled hash $H$ is large enough, the approximation error can be arbitrarily small. Therefore, $\vert \widehat{\mathsf{WAF3}} - \mathsf{WAF3} \vert$ can also be arbitrarily small, and naturally, the ordering of $\widehat{\mathsf{WAF3}}$ and $\mathsf{WAF3}$ will be very similar (or even identical). Therefore, the reason why $\widehat{\mathsf{WAF3}}$ can better identify over-squeezed edges is actually exactly the same as that of $\mathsf{WAF3}$ (i.e., appropriately modifying the weights of shared neighbors). This is essentially a trade-off: if the user can accept a larger error, then they can gain a greater speedup. In certain scenarios, the speedup provided by $\widehat{\mathsf{WAF3}}$ can be extremely useful. For example, if we want to find the edge with the minimum curvature in a large graph, we can first use $\widehat{\mathsf{WAF3}}$ to filter out the top 1% smallest edges, and then use precise $\mathsf{WAF3}$ calculations to select the exactly smallest one.
>
> We have also added an experiment that you may be interested in Appendix E.2. Specifically, we report the MOSR values of $\widehat{\mathsf{WAF3}}$ when $H$ takes 100, 1000, and 10000, respectively. We found that the $\mathsf{MOSR}$ values are quite stable for different orders of magnitude of $H$, and even when $H = 100$, the MOSR value is very close to that of the non-approximate ${\mathsf{WAF3}}$, and significantly lower than that of ${\mathsf{AF3}}$. We have only extracted the average results when the GNN is a GCN as shown below . For complete data, please refer to Table 14 in Section E.2 of the appendix of the paper.
>
> |  | MOSR_10 | MOSR_25 |
> | --- |---|---|
> | AF |  0.067  |  0.079  |
> | WAF | 0.036   |  0.045  |
> | Approx WAF (H=100) |  0.041  |  0.050  |
> | Approx WAF (H=1000)  | 0.041  | 0.048   |
> | Approx WAF (H=10000)  |  0.038  |  0.048  |

---

> ### Author Response · Authors · 2025-11-15
> **(2/N) About General Comment (2)**
>
> **General Comment (2)**
>
> > The paper lacks a large-scale, real-world evaluation demonstrating impact. A straightforward way to address this is to plug the efficient curvature into an existing rewiring pipeline and report results on a sparse OGB node property benchmark—e.g., ogbn-products or ogbn-papers100M—including both accuracy and compute (runtime/memory). This would substantiate scalability claims and show that the efficiency gains translate into practical benefits without undermining effectiveness.
>
> The focus of this paper is to reflect on and improve existing metrics, rather than to propose a new method. In particular, we would like to state the following two points:
>
> 1. **The progress of a metric is not necessarily related to the progress of methods based on that metric.** Graph rewiring algorithms are a good example of this. The first graph reconnection algorithm was proposed by Topping et al [1]. In their paper, they envisioned that graph rewiring algorithms could improve the performance of downstream tasks by alleviating over-squashing. However, a recent study [2] pointed out a surprising point: the effectiveness of graph rewiring does not actually stem from alleviating over-squashing, but from reducing outliers. This fundamentally overturns researchers' understanding of graph rewiring. Our paper proposes a definition of discrete edge curvature that is more effective and more efficient for detecting over-quashing edges, but this does not mean that it is more suitable for anomaly detection. This is just one example of graph rewiring, and we do not yet know whether other types of algorithms/methods also have similar mechanisms that have not yet been understood by researchers. Therefore, the performance of $\widehat{\mathsf{WAF3}}$ on specific downstream tasks cannot be used as a criterion for judging whether a metric is advanced.
>
> 2. For very large graphs (such as OGBN-Products or OGBN-Papers100M), even using $\widehat{\mathsf{WAF3}}$ as the curvature definition, executing discrete curvature-based algorithms similar to graph rewiring remains very difficult. **But, this difficulty lies not in calculating curvature, but in graph partitioning and the algorithm's own process.** We want to remind you that edge discrete curvature is a metric that depends only on the local graph (for an edge $u \sim v$, many definitions of discrete curvature only need to consider the subgraph consisting of $u$'s first-order neighbors and $u$'s first-order neighbors). Therefore, the computational efficiency of a single edge curvature is independent of the size of the entire graph. However, when the graph becomes extremely large, the situation becomes complicated because we cannot load the entire graph into memory simultaneously. In this case, we need graph partitioning techniques. If there happens to be a very large cluster in the graph, then in order to preserve complete first-order neighbors for curvature calculation for each edge, we need to partition the large graph into a considerable number of partially overlapping subgraphs. We have tried to extend the experiment in Figure 3 to 10^6 nodes or even more, but we found that the time consumed by graph partitioning would be tens of times that consumed by curvature calculation. On the other hand, even without considering the cost of calculating edge curvature and graph partitioning, graph rewiring on very large graphs is still not feasible. In classical graph rewiring algorithms (such as SDRF[1], RLEF[3] and FOSR[4], etc.), $\mathcal O(kn^2)$ edge curvatures need to be computed (where $n$ is the number of nodes, and $k$ is a constant related to the algorithm). Generally speaking, graph algorithms that are executed on very large graphs need to be carefully designed in terms of efficiency, but current curvature-based graph algorithms do not take this into account. Our work has cleared the obstacles to curvature calculation for scalable curvature-based graph algorithms, but much effort still needs to be put in, such as designing less complex graph partitioning and rewiring algorithms, but this is clearly beyond the scope of this paper.
>
> By the way, sampling-based methods are currently commonly used to perform graph algorithms on very large graphs; However, this is not feasible for curvature, because calculating curvature requires complete neighbor information, and there is still a considerable gap in this area for future research.
>
> **Reference**
>
> + [1] Understanding over-squashing and bottlenecks on graphs via curvature
> + [2] The Effectiveness of Curvature-Based Rewiring and the Role of Hyperparameters in GNNs Revisited
> + [3] Over- squashing in gnns through the lens of information contraction and graph expansion
> + [4] Fosr: First-order spectral rewiring for addressing oversquashing in gnns

---

> ### Author Response · Authors · 2025-11-16
> **(3/N) About Weakness 1~2**
>
> **Weakness 1 & 2**
>
> > Eqn (1) is not the MPNN model; that’s the GCN model. MPNN is a generalization of GCN. This is problematic since most of the results in over-squashing in the previous literature are for MPNNs, while in this paper, they’re only true for GCNs. This reduces the scope and impact of the proposed work and diminishes the results in Table 4, since the theoretical understanding is only available for GCN.
>
> > Similarly, multiple formal statements (Lemma 2, Theorem 5) say “Assume an L-layer MPNN as in (1).” But Eq. (1) is the symmetric-normalized GCN layer with ReLU; it is not a general MPNN. The proofs that rely on the exact matrix M therefore apply to GCN-style propagation; generalizations would require additional conditions on the aggregation kernels.
>
> We agree with your point; indeed, referring to the model described in the preparatory knowledge section of the paper as GCN, rather than MPNN, is more accurate. We have now thoroughly revised the paper to ensure rigorous descriptions. However, in addition, we would like to add the following three points regarding this issue.
>
> *1. Why don't we use an MPNN model similar to that in Topping et al.'s paper to complete the proof?*
>
> The fundamental reason is that the MPNN model used by Topping et al.[1] fails to provide information about the lower bound of the Jacobian matrix norm ($\mathsf{JacoNorm}$). Indeed, many papers on over-squashing report their theories based on MPNN rather than GCN. However, their assumptions regarding MPNN are similar to those of Topping et al. We invite you to open Topping et al.'s paper and observe their assumption (i) in Theorem 4 on page 5. You can see that they imposed nontrivial upper bounds on the gradient norms of the message and update functions in MPNN, but there are no nontrivial lower bounds. By assuming these nontrivial upper bounds, Topping et al. derived the global upper bound of the gradient for MPNN and linked this global upper bound to curvature, thus successfully explaining that a highly negative curvature is a sufficient condition for over-squashing. However, precisely because they did not assume a nontrivial lower bound for the gradient of MPNN, previous works all missed the point of discussing the necessity of over-squashing; that is, a low $\mathsf{JacoNorm}$ does not conversely mean that the curvature is highly negative. We use a specific GCN to complete the proof, which eliminates the need for additional lower bound assumptions for the gradient of MPNN, as GCN itself provides such a nontrivial lower bound.
>
> *2. How can the existing results in the paper be generalized to a more generalized MPNN model?*
>
> Let $\mathsf{MP}$ be the $\mathsf{MPNN}$ before the activation function, i.e., $\hat h^{(l)}_u = \mathsf{MP}(\{h_v^{(l)}|v\in \mathcal N(u)\})$ and $h^{(l+1)}_u = \mathsf{ACTIVE}(\hat h^{(l)}_u)$. When $\mathsf{MP}$ takes $\mathsf{GCN}$ as an instance, then $||\hat h^{(l)}_u / h_v^{(l)}|| =  \frac{|| W^{(l)} ||}{\sqrt{(d_u+1)(d_v+1)}}$. To extend the existing results, we can introduce the following additional assumption:
>
> Suppose there exists an $\alpha^{(l)}$ such that for every edge $u\sim v$ on the graph $\mathcal G$, we have: $||\hat h^{(l)}_u / h_v^{(l)}|| \geq \frac{\alpha^{(l)}}{\sqrt{(d_u+1)(d_v+1)}}$
>
> With this assumption, we can modify Lemma2, changing $|| \prod_{l=0}^{L-1} W^{(l)} ||$ in line 145 of the paper to $\prod_{l=0}^{L-1} \alpha^{(l)}$. This Lemma will then hold for all MPNNs that satisfy the assumption, not just GCNs. Furthermore, the other theorems in the paper will also apply to all MPNNs without requiring changes to their formulations.
>
> The core of our proofs is based on the derivation of the lower bound of $\mathsf{JacoNorm}$. The only difference is that before introducing additional assumptions, the lower bound was obtained from GCN itself, and after introducing additional assumptions, the lower bound is obtained with the help of these assumptions. There is no essential difference between the two.
>
> *3. If the theorem uses MPNN instead of GCN, will the conclusion change?*
>
> As discussed in the previous point, using MPNN to perform the proof does not provide any more insight than GCN. Furthermore, experimentally, we found that GCN is by comparison the least likely model to overlook squeeze edges (see Observation 2 in the paper), and we proved that even this best-performing model in practice is theoretically flawed. Therefore, we have considerable confidence that other models (such as GAT and SAGE) also have theoretical flaws. In summary, all the conclusions (observations) in our paper remain unchanged.
>
> **Reference**
> + [1] Understanding over-squashing and bottlenecks on graphs via curvature

---

> ### Author Response · Authors · 2025-11-17
> **(4/N) About Weakness 3**
>
> **Weakness 3**
> > The last part of Section 4 lacks clear intuition on how over-squashing relates to bottlenecks and how this differs from “bridges”. Prior theory (e.g., Topping et al.) does not imply that removing bottlenecks eliminates over-squashing, yet the paper does not provide a concrete narrative or example to reconcile this. In particular, the manuscript does not illustrate what an intra-cluster over-squashed edge looks like (low betweenness, still over-squashed) or explain the mechanisms that make it over-squashed despite no obvious bottleneck. This gap makes the conceptual contribution harder to follow and weakens the takeaways of Section 4.
>
> We divide our rebuttal to this comment into the following four parts.
>
> *1. how over-squashing relates to bottlenecks and how this differs from “bridges”?*
>
> We believe that reviewers may have some misunderstandings regarding the statements in lines 308 to 314 of the paper, and we would like to clarify this. We believe that over-squashing is one of the core challenges in graph learning, and **bottlenecks are the topological factors that lead to over-squashing**. On this point, our view is consistent with that of Topping et al.
>
> > "... bottleneck, defined as those topological properties in the graph leading to over-squashing." --- Topping et al.
>
> However, we believe that our previous literature has not rigorously clarified the relationship between bottlenecks and bridges. In our paper, we argue that **bridges are a type of bottleneck, but not the only one**. Because bridges occur between clusters, and there is another type of bottleneck within a cluster that does not manifest as a bridge.
>
> *2. What were the viewpoints of previous theories (e.g., Topping et al.)?*  --- Topping et al.
>
> In fact, Topping et al. explicitly stated that removing the bottleneck would alleviate over-squashing. See page 6 of Topping et al.'s paper for details:
>
> > "we assume that graph rewiring attempts to produce a new graph $G^\prime = (V, E^\prime)$ with a different edge structure that **reduces the bottleneck and hence potentially alleviates the over-squashing of information.**"
>
> We would like to emphasize that we agree with this viewpoint. However, the graph rewiring algorithm proposed by Topping et al. can only detect and remove bridges as a bottleneck. It is powerless against another type of bottleneck—the bottleneck within a cluster.
>
> *3. What dose the intra-cluster over-squashed edge looks like?*
>
> We invite you to observe Figure 1 in our paper --- the counterexample we present. For the source node $\mathsf s$ and the target node $\mathsf t$, they share all their first-order neighbors ($\mathcal N_1$), so they are obviously in the same cluster. However, the information of each first-order neighbor node is diluted by $m$ unrelated nodes. Therefore, only a small amount of information from $\mathsf s$ reaches $\mathsf t$, which manifests as over-squashing. At this point, we can consider that there exists a bottleneck between $\mathsf s$ and $\mathsf t$ that does not manifest as a bridge.
>
> *4. What mechanisms make it (intra-cluster over-squashed edge) over-squashed despite no obvious bottleneck?*
>
> After our explanation above, you should now understand that the question is incorrect. inter-cluster over-squashing edges are also caused by bottlenecks, but not bridge-shaped bottlenecks. Instead, they are bottlenecks that resemble the edge (s,t) in Figure 1 (we haven't given it a specific name yet).

---

> ### Author Response · Authors · 2025-11-17
> **(5/N) About Weakness 4~5**
>
> **Weakness 4**
>
> > The choice of the Forman-3 curvature for the extension seems arbitrary. Indeed, the balance Forman w/o 4-cycle and the Jost-Liu Forman have the same complexity as the Augmented Forman-3 according to Table 1. However, those are not compared in Table 2, and we don’t know if these other two computationally efficient metrics are better or worse than the Forman-3 curvature.
>
> We chose AF3 for extension for the following two reasons: 1. AF3 is used far more frequently in the paper than Balance Forman with 4-cycle and the Jost-Liu Forman. But more importantly: 2. Based on Equation 6 in the main text, it is natural to generalize AF3 to effectively alleviate situations similar to the counterexamples. However, how to perform a similar extension of Balance Forman with 4-cycle and the Jost-Liu Forman remains unknown. We now supplement our report with the results of Balance Forman with 4-cycle and the Jost-Liu Forman on MOSR (see Appendix E.3), and we have extracted their average performance as follows. Overall, Balance Forman with 4-cycle and the Jost-Liu Forman perform slightly worse than to AF3, and significantly worse than our improved WAF3.
>
> | Model = GCN | MOSR_10 | MOSR_25|
> | --- | --- | --- |
> | Balance Forman with 4-cycle| 0.073 | 0.089 |
> | Jost-Liu Forman | 0.085 | 0.088 |
> | AF3 | 0.067 | 0.079 |
> | WAF3 | 0.036 | 0.045 |
>
> **Weakness 5**
> > I understood the reasoning behind Figure 4, but I feel it’s also important to compare the efficient version of the proposed curvature in terms of MOSR, as in Table 4. This new version might be more efficient, but the main point of this paper is to avoid missing the over-squashed edges where the curvature is positive, and if this is not the case for the efficient formulation, why should one go with this alternative? If the important point is the ordering (I agree with the authors on this), why should one choose this approach over something that achieves the correct rank, while being perhaps more efficient?
>
> We have now added such an experiment, located in Appendix E.2, which we also mentioned and quoted the relevant results in our reply **(1/N) About General Comment (1)**, so we will not repeat it here. The experimental results show that the MOSR value of the MinHash-based approximation WAF3 is only slightly degraded relative to the exact WAF3, but is still significantly better than AF3. Therefore, the MinHash-based approximation WAF3 strikes a good balance between efficiency and performance—it not only significantly speeds up the computation, but also maintains a low MOSR value and a high Kendall-Tau-b ranking similarity.

---

> ### Author Response · Authors · 2025-11-17
> **(6/N) Minor comments & Questions**
>
> **Minor Comments**
>
> > Line 58: “This theorem…”, which theorem? At this point in the paper, it’s not clear this is supported by a theorem.
>
> This refers to Thm 4. We have revised the original text to ensure clarity of reference.
>
> > Please define acronyms the first time they’re used, e.g., MOSR.
>
> We have now added a full name on line 62.
>
> > It seems like Table 3 has not been referenced in the paper.
>
> Table 3 has now been properly referenced at row 300.
>
> > Figure 2 is violating the margins.
>
> We have adjusted the size and position of Figure 2 to ensure it conforms to the margins.
>
> > Typo in Theorem 5? There dose not.
>
> There is indeed a Thm5, it's in lines 358 to 361.
>
> > Could you include the statistics of the datasets? I think this is important to put things in context.
>
> Of course. Now we have added a dataset statistic in Appendix E.1.
>
> **Questions**
>
> > What is \rho in Lemma 2? This also appears in Appendix A.1, but I’m not sure what this is.
>
> $\rho$ is introduced in Assumption 1.
>
> > Assumption 1 is not clear. What does the paper mean by all paths in the computation graph? What’s the computation graph in this case?
>
> Assumption 1 is often used to simplify the analysis of ReLu MPNN, and we borrow the description from paper [1] in this paper.
> In simple terms, a computation graph is a graph of gradient dependencies. Take a two-layer MPNN as an example, assuming $u \sim v$ and $v \sim w$. If we want to backpropagate $h_w^{(2)}$, then the gradient of $h_w^{(2)}$ depends on the gradient of $h_v^{(1)}$, which in turn depends on the gradient of $h_u^{(0)}$. Therefore, $h_u^{(0)} \rightarrow h_v^{(1)} \rightarrow h_w^{(2)}$ is a path in the computation graph of $h_w^{(2)}$. Recall that the ReLU function filters out a portion of the input (i.e., the part greater than zero) from the whole input to continue propagating. Since the number of ReLU functions traversed along each information flow path from the input features to the final output features is the same, Therefore, we can assume that, in terms of expectation, a certain proportion ($\rho$) of information from each path is preserved. The essence of this assumption is to transform the nonlinear ReLU function into a linear scaling effect on features by considering the expectation. Many famous GNN papers have used this assumption to assist in analysis, such as JKNet [2].
>
>
> > The notation in Definition 3 is not clear. $\mathcal N_1$ and $\mathcal N_2$ are the 1-hop and 2-hop neighborhoods of whom?
>
> In the counterexample diagram we provided, $\mathcal N_1$ and $\mathcal N_2$ are both first-order neighbors and second-order neighbors of the source node $\mathsf s$, and also first-order neighbors and second-order neighbors of the target node $\mathsf t$.
>
> > What’s the number of layers L for the experiments in Table 2?
>
> To ensure the experiments were conducted in a realistic graph learning context, we performed 200 hyperparameter searches for each GNN on each dataset and used the optimal combination of hyperparameters when computing MOSR. For specific values ​​of L, please refer to Appendix C and Tables 7, 8, and 9, where all hyperparameters are listed in detail.
>
> > Could the authors provide the same results as in Table 2 for the balance Forman w/o 4-cycle and the Jost-Liu Forman? These two curvature metrics are also computationally efficient.
>
> Sure, You can now see this experiment in Appendix E.3 and Table 15.
>
> > In Appendix A.1, Lemma 5 is used to get the bound, but I didn’t find this Lemma. Is it Lemma 7? Even if this is the case, I’m not quite sure how to get the bound in the first Equation on page 14 (please label the equations). Why is greater than or equal to? I’m confused at that point.
>
> Yes, this refers to Lemma 7, which we have corrected in the paper. The derivation from line 701 to line 705 is as follows: First, take the norm of both sides of line 701 (any norm definition will do), and then extract all positive constants outside the norm sign for the right side of the equation. Then apply Lemma 7 to get the result. Note that we followed the proof method and notation conventions in pages 17 to 18 of the paper [1], thus omitting the expectation sign on the left side of the inequality.
>
>
> **Reference**
> + [1] On Over-Squashing in Message Passing Neural Networks: The Impact of Width, Depth, and Topology
> + [2] Representation Learning on Graphs with Jumping Knowledge Networks

---

> > ### Comment · Reviewer_jUf2 · 2025-11-28
> >
> > I thank the authors for the answers. I still believe this efficient curvature metric should be tested on large-scale datasets to see the utility of rewiring in GNNs in general. However, I agree with the authors that this is a broader problem, and it might be more appropriate to address it in future work. In any case, I think this is a work that helps the community to progress on the understanding of over-squashing and its technical implications, so I'll recommend this paper for acceptance, and I'll increase my score (once I can do that). I cannot edit my score at this moment.
> >
> > Two last comments:
> >
> > - Please keep answers concise. It was somehow easier to go through the paper again than to go through the answers.
> > - There are still some remaining issues, like the mention of MPNN on page 20. Please be sure to fix those for the camera-ready in case the paper is accepted.

---

### Official Review · Reviewer_5cz2 · 2025-10-31

**Soundness:** 4
**Presentation:** 4
**Contribution:** 4
**Rating:** 10
**Confidence:** 4

**Summary:**

The authors address the problem of over-squashing in GNNs. In this context, they investigate the following point: high negative curvature is a sufficient but not a necessary condition for over-squashing. In addition, they develop an approximation algorithm for the method Weighted Augmented Forman-3 Curvature.

**Strengths:**

- The paper is well organized and written
- The authors address a key issue about GNNs: over-squashing.
- The proposed method Weighted Augmented Forman-3 Curvature is detailed and reproducible.
- Experiments are well conducted.

**Weaknesses:**

The authors should indicate more information between over-squashing and over-smoothing.
Missing references:
A. Arnaiz-Rodríguez, F. Errica, “Oversmoothing, Oversquashing, Heterophily, Long-Range, and More: Demystifying Common Beliefs in Graph Machine Learning”, Preprint, May 2025.
Y. Liu, et al., "CurvDrop: A Ricci curvature based approach to prevent graph neural networks from over-smoothing and over-squashing",  ACM Web Conference 2023, pages 221-230, 2023.

**Questions:**

None

---

> ### Author Response · Authors · 2025-11-19
>
> Thank you for your comment. Here is our detailed response.
>
> > The authors should indicate more information between over-squashing and over-smoothing.
>
> We have now added a new section, “E.4: Over-squashing and over-smoothing,” to the appendix to explore the relationship between these two concepts in detail. We have also pasted the contents of that section below for your reading.
>
> Over-smoothing and over-squashing are both significant challenges in designing GNNs, and they are distinct but also deeply connected. On the one hand, paper [1] clearly points out the obvious difference between the two concepts: oversmoothing refers to the problem of node features becoming too smooth when the number of layers in a GNN is too large; over-squashing refers to the problem of long-range information not being effectively utilized due to bottleneck structures in the graph. On the other hand, papers [4, 6] provide a unified perspective on over-smoothing and over-squashing from the perspectives of spectral theory and curvature, respectively. Specifically, [4] argues that excessively large/small spectral gaps lead to over-smoothing/over-squashing; while [6] argues that highly positive/negative Ollivier-Ricci curvatures lead to over-smoothing/over-squashing. Furthermore, [5] points out the consistency between over-smoothing and over-squashing in causing gradient vanishing.
>
> As for the methodological level, many works have attempted to alleviate both challenges simultaneously. Among them, [2] proposed a dropout and sampling method based on Ollivier-Ricci curvature; [3] proposed a track propagation mechanism to avoid information "hybridization" from different sources; and [7] designed a graph rewiring method based on Augmented Forman-Ricci Curvature.
>
> Although our paper focuses on the relationship between curvature and over-squashing, given the central role of curvature in understanding and solving over-squashing problems, we have reason to believe that our work will also briefly inspire researchers dedicated to studying over-smoothing problems. In particular, we can similarly consider whether the statement in [6] that "highly positive curvature leads to over-smoothing" is sufficient or necessary. We will leave the discussion here to future work.
>
> **Reference**
> + [1] Oversmoothing, Oversquashing, Heterophily, Long-Range, and More: Demystifying Common Beliefs in Graph Machine Learning
> + [2] CurvDrop: A Ricci curvature based approach to prevent graph neural networks from over-smoothing and over-squashing
> + [3] Multi-Track Message Passing: Tackling Oversmoothing and Oversquashing in Graph Learning via Preventing Heterophily Mixing
> + [4] On the Trade-off between Over-smoothing and Over-squashing in Deep Graph Neural Networks
> + [5] On Vanishing Gradients, Over-Smoothing, and Over-Squashing in GNNs: Bridging Recurrent and Graph Learning
> + [6] Revisiting Over-smoothing and Over-squashing Using Ollivier-Ricci Curvature
> + [7] Mitigating Over-Smoothing and Over-Squashing using Augmentations of Forman-Ricci Curvature

---

### Official Review · Reviewer_hwjY · 2025-11-01

**Soundness:** 3
**Presentation:** 3
**Contribution:** 3
**Rating:** 8
**Confidence:** 4

**Summary:**

In this paper, the authors question the connection of edge curvature to over-squashing. These edges are identified by a discrete notion of curvature (which is specifically adapted for graphs) that assigns a curvature value to each edge. They specifically claim that high negative curvature is a sufficient but not a necessary condition for over-squashing. To show that, they create counterexamples where some of the edges are squashed while the curvature remains positive. Since the Ollivier–Ricci curvature, one of the most commonly used discrete curvature measure, fails to detect a considerable percentage of over-squashed edges, the authors propose the Weighted Augmented Forman-3 Curvature (WAF3) to improve the detection of over-squashed edges. Finally, a new approximated WAF is presented that is able to handle very large graphs.

**Strengths:**

- studies the over-squashing phenomenon in GNNs
- shows that over-squashed edges may not always be detected by curvature
- newly introduced metric for measuring over-squashed edges ratio by curvature-based criteria

**Weaknesses:**

- Counterexamples may not be representative of real-world graphs (e.g., citation, molecule, social). The authors should provide synthetic constructions with empirical evidence that similar topology/feature interactions occur in practical datasets.
- Weighted Jaccard and MinHash are algorithmic conveniences; the transformation may sacrifice geometric interpretability
- Proposed WAF3 correlates with or mitigates real over-squashing beyond proxy metrics

**Questions:**

- How robust is MOSR under different operational definitions of over-squashing?
- Do analogous failure cases appear in real-world graphs (social, citation, molecular)?
- Does WAF3-based rewiring or edge weighting actually alleviate over-squashing and improve downstream accuracy?

Typos
- line 155: Let s the soruce node, -> source

---

> ### Author Response · Authors · 2025-11-23
> **(1/N) About Weakness 1&2**
>
> We appreciate the reviewers' efforts in providing professional feedback, and below is our detailed response.
>
> **Weakness 1**
> > Counterexamples may not be representative of real-world graphs (e.g., citation, molecule, social). The authors should provide synthetic constructions with empirical evidence that similar topology/feature interactions occur in practical datasets.
>
> As we demonstrate in Table 3 and Observations 4 and 5, we argue that graph structure bottlenecks leading to over-squashing exist not only between clusters but also within clusters, and that existing discrete curvatures fail to be a necessary and sufficient condition for identifying over-squashing precisely because they ignore intra-cluster bottlenecks. Our counterexample, however, represents an extreme case of an intra-cluster bottleneck (since s and t share all their first-order neighbors), and is therefore representative of the real world.
>
> Specifically, let $\mathcal E_{\mathsf{igno}}$ / $\mathcal E_{\mathsf{iden}}$ represent the sets of over-squashed edges that are ignored / identified, and let $t_e$ and $s_e$ represent the nodes at both ends of an edge $e$. $\triangle_e = |\mathcal{N}(t_e)\cap\mathcal{N}(t_s)|$ is the number of triangles formed by $e$ and $d_e = \mathsf{Mean}(deg(v)|v\in \mathcal{N}(t_e)\cap\mathcal{N}(t_s))$ is the average degree of the third vertex of these triangles.
>
> Based on the counterexamples we provide, we expect ${\mathsf{Mean}(\Delta_e | e\in \mathcal E_{\mathsf{igno}})} > {\mathsf{Mean}(\Delta_e | e\in \mathcal E_{\mathsf{iden}})} $  and  ${\mathsf{Mean}(d_e | e\in \mathcal E_{\mathsf{igno}})} > {\mathsf{Mean}(d_e | e\in \mathcal E_{\mathsf{iden}})} $  (for the reasons, please refer to our intuitive explanation of the counterexamples in Chapter 3). We have further supplemented this with experiments to demonstrate that the phenomenon does indeed exist:
>
> |   | $ \Delta_{igno} $ | $ \Delta_{iden} $ | $ d_{igno} $ | $ d_{iden} $ |
> | --- | --- | --- | --- | --- |
> | Cora | 3.1  | 0.4  | 5.2 | 2.2 |
> | Computer | 6.6| 1.7 | 9.3 | 5.2 |
>
> where:
> $\Delta_{igno} := {\mathsf{Mean}(\Delta_e | e\in \mathcal E_{\mathsf{igno}})} $, $\Delta_{iden}= {\mathsf{Mean}(\Delta_e | e\in \mathcal E_{\mathsf{iden}})}$, $d_{igno} := {\mathsf{Mean}(d_e | e\in \mathcal E_{\mathsf{igno}})} $ and  $d_{iden} := {\mathsf{Mean}(d_e | e\in \mathcal E_{\mathsf{iden}})} $
>
> Therefore, we can say that this counterexample does exist in graphs in the real world.
>
> **Weakness2**
> > Weighted Jaccard and MinHash are algorithmic conveniences; the transformation may sacrifice geometric interpretability
>
> The MinHash-based approximation is an unbiased statistic, and when the number of hash functions $H$ is sufficiently large, the expected value of the approximation error can be arbitrarily small. Therefore, theoretically, the MinHash-based WAF3 has virtually the same geometric interpretation as the exact WAF3.

---

> ### Author Response · Authors · 2025-11-23
> **(2/N) About Weakness 3 & Questions**
>
> **Weakness 3 & Question 3**
>
> > Proposed WAF3 correlates with or mitigates real over-squashing beyond proxy metrics
>
> > Does WAF3-based rewiring or edge weighting actually alleviate over-squashing and improve downstream accuracy?
>
> In Appendix D.3, we analyzed the performance of WAF3 and its approximations on downstream tasks, including graph rewiring methods and end-to-end depth graph models. The results show that our proposed curvature outperforms existing algorithms in the vast majority of cases. But besides that, we want to emphasize that: our paper focuses on improving the definition of curvature rather than enhancing the performance of downstream tasks. And **the progress of a metric is not necessarily related to the progress of methods based on that metric.** Graph rewiring algorithms are a good example of this. The first graph rewiring algorithm was proposed by Topping et al [1]. In their paper, they envisioned that graph rewiring algorithms could improve the performance of downstream tasks by alleviating over-squashing. However, a recent study [2] pointed out a surprising point: the effectiveness of graph rewiring does not actually stem from alleviating over-squashing, but from reducing outliers. This fundamentally overturns researchers' understanding of graph rewiring. Our paper proposes a definition of discrete edge curvature that is more effective and more efficient for detecting over-quashing edges, but this does not mean that it is more suitable for anomaly detection. This is just one example of graph rewiring, and we do not yet know whether other types of algorithms/methods also have similar mechanisms that have not yet been understood by researchers.
>
> **Question 1**
>
> > How robust is MOSR under different operational definitions of over-squashing?
>
> Table 2 in the main text presents the MOSR values ​​for three different curvature definitions under the same settings. We have also added results for two additional curvature definitions in Table 15 of the appendix. We found that the results for Augmented Forman-3, Balance Forman w/o 4-cycle, and Jost-Liu Forman curvatures are very close and superior to the remaining two curvatures. Therefore, no single curvature is significantly better than all other existing curvatures, and thus MOSR can robustly reflect the ability of existing curvatures to detect over-squashing phenomena.
>
> **Question 2**
>
> > Do analogous failure cases appear in real-world graphs (social, citation, molecular)?
>
> Our experiments encompassed 21 of the most common node-level datasets, including: citation networks (e.g., Cora, MOSR_25=0.103), hyperlink networks (e.g., Chameleon, MOSR_25=0.643), etc. Ollivier-Ricci curvature showed significant failure on these datasets. In addition, we supplemented our experiments with the following two datasets:
>
>
> | | MOSR_10 | MOSR_25|
> | --- | --- | --- |
> | BolgCatalog (Social Network) | 0.112 | 0.158 |
> | PPI (Molecula Network) | 0.157  | 0.206 |
>
> Therefore, the phenomenon of failing to identify over-squashing edges is widespread across different types of datasets.
>
>
>
> [1] Understanding over-squashing and bottlenecks on graphs via curvature
>
> [2] The Effectiveness of Curvature-Based Rewiring and the Role of Hyperparameters in GNNs Revisited

---

> > ### Comment · Reviewer_hwjY · 2025-11-23
> >
> > I thank the authors for their answers, extensive analysis and specific examples to address my concerns. I keep my positive score and strongly encourage to integrate the answers into the next version of the paper (hopefully the camera-ready).

---

### Official Review · Reviewer_fzn1 · 2025-11-03

**Soundness:** 3
**Presentation:** 4
**Contribution:** 3
**Rating:** 6
**Confidence:** 2

**Summary:**

This paper considers the connection between the oversquashing problem and negatively curved edges (quantified using a variety of different notions of discrete curvature). They show, contrary to popular belief, that negative curvature is merely a sufficient, rather than a necessary, condition for oversquashing to occur and show empirically that many oversquashed edges (quantified via jacobian norms) are "missed" by checking the curvature. To remedy this, the authors introduce a novel version of discrete curvature, as well as a fast approximation algorithm.

**Strengths:**

Oversquashing is a well-known, highly studied problem in the training of GNNs. This paper lends a new perspective showing that the prevailing wisdom, oversquashing <==> negative curvature, is inadequate. They also show experimentally which of the common GNN architecture are prone to oversquashing.

Empiricial results are supported by theoretical analysis and novel definition which provide a new framework for thinking about oversquashing

**Weaknesses:**

Assumption 1 comes out of no where and is hard to understand. More discussion and motivation should be given.

The definition of $\mu_u^\alpha$, is unclear

**Questions:**

Are there any viable ways to understand oversquashing which are not curvature derived?

---

> ### Author Response · Authors · 2025-11-19
>
> We appreciate the reviewers' hard work, and here is our response.
>
>
> **Weakness 1**
> > Assumption 1 comes out of no where and is hard to understand. More discussion and motivation should be given.
>
> Assumption 1 is used to simplify the analysis of ReLU MPNN, and we borrow the description from paper [1] in this paper.
> In simple terms, the ''computation graph'' mentioned in the assumption is a graph of gradient dependencies. Take a two-layer MPNN as an example, assuming $u \sim v$ and $v \sim w$. If we want to backpropagate $h_w^{(2)}$, then the gradient of $h_w^{(2)}$ depends on the gradient of $h_v^{(1)}$, which in turn depends on the gradient of $h_u^{(0)}$. Therefore, $h_u^{(0)} \rightarrow h_v^{(1)} \rightarrow h_w^{(2)}$ is a path in the computation graph of $h_w^{(2)}$. Recall that the ReLU function filters out a portion of the input (i.e., the part greater than zero) from the whole input to continue propagating. Since the number of ReLU functions traversed along each information flow path from the input features to the final output features is the same, Therefore, we can assume that, in terms of expectation, a certain proportion ($\rho$) of information from each path is preserved. The essence of this assumption is to transform the nonlinear ReLU function into a linear scaling effect on features by considering the expectation. Many famous GNN papers have used this assumption to assist in analysis, such as JKNet [2].
>
> **Weakness 2**
> > The definition of $\mu_u^\alpha$, is unclear
>
> $\mu_u^\alpha$ is the uniform distribution of the first-order neighbors of u with restart probability $\alpha$. Specifically, suppose $u$ and $v$ are any nodes in a graph without self-loops $\mathcal G$, then $\mu_u^\alpha$ is such a function that: when $v = u$, $\mu_u^\alpha(v) = \alpha$, otherwise, $\mu_u^\alpha(v) = \frac{1 - \alpha}{d_u}$. We can consider $\mu_u^\alpha$ as a random walk function. When a random walk path reaches $u$, it will move to any of its first-order neighbors with equal probability at the next time step, while there is a probability $\alpha$ that will not move (this is similar to a restart in a random walk). $\mu_u^\alpha$ is common in discrete curvatures based on optimal transport, such as the Ollivier-Ricci curvature and the Lin-Lu-Yau curvature.
>
> **Question 1**
> > Are there any viable ways to understand oversquashing which are not curvature derived?
>
> From the perspective of solving over-squashing, there are indeed some works that do not rely on curvature. According to the research results in the latest survey [3], such as introducing additional fully connected layers [4], layer-dependent rewiring [5], and rewiring based on total resistance [6]. However, from the perspective of theoretically understanding over-squashing, the curvature perspective is dominant. A few exceptions are: explaining over-squashing from the perspective of effective resistance [6, 7] and from the perspective of spectral gap [8]. However, both perspectives acknowledge that they are very closely related to curvature, for example:
>
> > "In some ways, the effective resistance and Balanced Forman curvature of an edge are similar, as both measure how con- nected the endpoints are. However, our analysis generalizes the previous bound in several important ways." --- Black et. al ([6])
>
> > "a positive lower bound on the curvature gives us a control on $h_G$ and hence on the spectral gap of the graph" --- Topping et. al (8)
>
> Therefore, although our work is a comprehensive reflection on over-suqashing based on curvature, its influence will be extended to the entire field due to the central role of the curvature perspective.
>
> **Reference**
> + [1] On Over-Squashing in Message Passing Neural Networks: The Impact of Width, Depth, and Topology
> + [2] Representation Learning on Graphs with Jumping Knowledge Networks
> + [3] Over-squashing in Graph Neural Networks: A comprehensive survey
> + [4] ON THE BOTTLENECK OF GRAPH NEURAL NETWORKS AND ITS PRACTICAL IMPLICATIONS
> + [5] DRew: Dynamically rewired message passing with delay, in: International Conference on Machine Learning
> + [6] Understanding oversquashing in gnns through the lens of effective resistance
> + [7] DiffWire: Inductive Graph Rewiring via the Lovász Bound
> + [8] UNDERSTANDING OVER-SQUASHING AND BOTTLENECKS ON GRAPHS VIA CURVATURE

---

### Comment · Area_Chair_TQY9 · 2025-11-23
**Next Steps Following Authors’ Rebuttal: Review Rebuttal and Participate in Discussion**

Dear Reviewers,

Thank you very much for your thoughtful evaluations of this paper.

Now that the authors have submitted their rebuttal, I kindly ask you to take the following steps (if you have not done so already):

- Read the other reviews as well as the authors’ response.
- Consider whether the rebuttal and additional comments affect your assessment of the paper.
- Engage in interactive discussion with the authors **before November 25**, encouraging a dynamic exchange rather than a one-sided rebuttal.

Your contributions at this stage are essential for forming a well-informed final decision. I therefore ask that you reassess your views in light of the authors’ responses and the broader discussion among reviewers.

I am happy to join and support the discussions between you and the authors. Please feel free to share your thoughts and participate actively in the discussion.

Thank you once again for your service to ICLR 2026.

Best regards,

 AC

---

### Author Response · Authors · 2025-12-03
**Summary**

Our paper aims to highlight a cognitive bias prevalent in the graph learning community regarding the use of discrete curvature to understand and address the over-squashing problem, while also proposing more reasonable and efficient metrics to alleviate this challenge. Our contribution is both critical and constructive.

Our initial review scores were **(6, 8, 10, 4, 6)**. During the rebuttal phase, we addressed every concern and issue raised by the reviewers in a detailed, point-by-point manner. Subsequently, two reviewers provided feedback on our rebuttal. We are pleased to report that we ultimately reached consensus with all reviewers:

+ Reviewer **TQY9** acknowledged that our paper introduces a new perspective challenging conventional wisdom and provides a novel framework for understanding over-squashing.
+ Reviewer **hwjY** confirmed that our rebuttal resolved their concerns and maintained a positive score (**8**).
+ Reviewer **5cz2** commended the paper for being well-written, reproducible, and featuring well-executed experiments.
+ Reviewer **jUf2**, after considering our rebuttal, recognized that our work advances the community's understanding of and solutions to the over-squashing problem, and expressed willingness to raise their score and recommend acceptance of the paper.
+ Reviewer **QN1W** found the paper to be original, with solid technical quality and clear presentation.

During the rebuttal period, we further updated the paper to enhance its quality, specifically by:

+ Revising the statements of certain lemmas and theorems to better align with their corresponding formulas.
+ Adding statistical details for all datasets used in the paper (Table 13).
+ Introducing two new experiments (Tables 14 and 15).
+ Correcting multiple typos and adjusting the margin of certain figures.
+ Expanding the discussion on the relationship between over-smoothing and over-squashing.

In the future, we will continue refining the paper based on reviewers' feedback by adding further explanations and experiments.

Finally, we would like to express our sincere gratitude once again to the Area Chair and all reviewers for their valuable time and insightful suggestions.

---

### Meta-Review · Area_Chair_FXMP · 2026-01-06

**Summary:**

In their paper, the authors critically re-examine the widely adopted practice of using discrete curvature as a proxy for detecting over-squashing in graph neural networks. The central contribution is a combined theoretical and empirical demonstration that while highly negative curvature is sufficient for over-squashing, it is not necessary, thereby challenging the prevailing view of curvature as a "gold standard" diagnostic. The authors formalize this gap through a constructive counterexample, introduce a new evaluation metric to quantify missed over-squashing, and propose Weighted Augmented Forman-3 (WAF3) curvature, along with an efficient approximation algorithm to improve the detection of over-squashed edges. Overall, this is a solid paper with technically sound contributions.

(1) *Novelty and Technical Soundness*: The paper proves that discrete curvature, despite its widespread use, is fundamentally incapable of fully characterizing over-squashing. Through a carefully constructed family of graphs, the authors show that edges can suffer severe Jacobian-based information collapse while maintaining positive curvature under eight widely used definitions, including Ollivier–Ricci and Balanced Forman curvature. Additionally, the proposed WAF3 method to detect over-squashed edges introduces degree-based weighting in an intuitive manner, retains low computational complexity, and is shown to eliminate the constructed counterexample under mild conditions. Most reviewers agreed that the theoretical arguments are technically sound, and that the clarified assumptions and scope in the rebuttal further solidified the analysis.

(2) *Empirical Evaluation and Practical Impact*: The empirical evaluation was thorough and well aligned with the paper's goals. In particular, Reviewer jUf2 gave positive feedback for the introduction of the Missed Over-Squashing Ratio (MOSR) as a new metric to quantify failures of curvature-based diagnostics and Reviewer 5cz2 found the experiments across datasets, architectures, and curvature notions convincing in demonstrating that commonly used measures (e.g., Ollivier–Ricci curvature) can miss a nontrivial fraction of over-squashed edges. Furthermore, the proposed WAF3 and its corresponding approximation algorithm consistently outperforms curvature-based methods in preserving detection behavior with significant speedups.

In summary, the paper provides a meaningful correction to current practice and offers a principled alternative for diagnosing over-squashing in GNNs. Therefore, I recommend acceptance.

**Reviewer Concerns:**

Please refer to the summary.

**Reviewer Scores:**

Please refer to the summary.

---

### Decision · Program_Chairs · 2026-01-26

Accept (Poster)